# Insertion sequence transposition inactivates CRISPR-Cas immunity

Yong Sheng [1,4], Hengyu Wang[1,4], Yixin Ou[1,2], Yingying Wu[1,3], Wei Ding[1], Meifeng Tao[1,2], Shuangjun Lin [1,2], Zixin Deng [1,2] ✉, Linquan Bai [1] ✉ & Qianjin Kang [1,2] ✉

CRISPR-Cas immunity systems safeguard prokaryotic genomes by inhibiting the invasion of mobile genetic elements. Here, we screened prokaryotic genomic sequences and identified multiple natural transpositions of insertion sequences (ISs) into *cas* genes, thus inactivating CRISPR-Cas defenses. We then generated an IS-trapping system, using *Escherichia coli* strains with various ISs and an inducible *cas* nuclease, to monitor IS insertions into *cas* genes following the induction of double-strand DNA breakage as a physiological host stress. We identified multiple events mediated by different ISs, especially IS1 and IS10, displaying substantial relaxed target specificity. IS transposition into *cas* was maintained in the presence of DNA repair machinery, and transposition into other host defense systems was also detected. Our findings highlight the potential of ISs to counter CRISPR activity, thus increasing bacterial susceptibility to foreign DNA invasion.

The acquisition of foreign DNA by horizontal transfer, such as genes for antibiotic resistance, can increase bacterial fitness[1,2]. However, prokaryotes also have defense mechanisms to counter the invasion of viruses and other genetic parasites[3,4]. The clustered regularly interspaced short palindromic repeat (CRISPR)/Cas systems are a dominant defense mechanism that protect prokaryotes against invasive genetic elements through the use of specific guide RNAs (crRNAs) generated from the CRISPR array, a bank of DNA sequences inserted in the host genome and derived from foreign genetic material[5,6]. crRNAs "program" the CRISPR-associated (Cas) protein to bind and cleave target sequences complementary to the crRNA sequence[7]. On the basis of the signature Cas effectors and mechanistic properties, CRISPR/Cas systems are currently classified into two classes and six types[8]. Type II-A SpCas9 endonuclease from *Streptococcus pyogenes* harbors two nuclease domains, RuvC and HNH, which can cleave the non-target strand and the target strand, respectively[9,10]. Innate restriction-modification (RM) immune systems[11] also limit genetic parasitism as

do other prokaryotic defense elements, such as abortive infection systems[12] and toxin-antitoxin systems[13].

However, despite immune systems, insertion sequences (ISs) and other mobile genetic elements (MGEs) still broadly mediate horizontal gene transfer across species[14,15]. ISs, the highly prevalent MGEs in nature, are genetically compact, flanked by inverted terminal repeats, and generally phenotypically cryptic mobile elements that encode only transposases to facilitate their movement, which typically results in target site duplications (TSDs) of variable length flanking the insertion sites[16,17]. Due to their random transposition and the potential for homologous recombination between two identical IS copies, ISs can sometimes have deleterious effects on the host[18,19]. However, ISs, particularly in composite transposons, can also provide some survival advantages to the host, including through modulating metabolism[20], facilitating DNA repair[21], and enhancing virulence and antimicrobial resistance[22,23]. Therefore, IS transposition into the host can contribute to mutual survival.

[1]State Key Laboratory of Microbial Metabolism, Joint International Research Laboratory of Metabolic & Developmental Sciences, and School of Life Sciences & Biotechnology, Shanghai Jiao Tong University, 200240 Shanghai, P. R. China. [2]Haihe Laboratory of Synthetic Biology, 300308 Tianjin, P. R. China. [3]National Engineering Research Center of Edible Fungi, Key Laboratory of Applied Mycological Resources and Utilization (South), Ministry of Agriculture and Rural Affairs, Institute of Edible Fungi, Shanghai Academy of Agricultural Sciences, 201403 Shanghai, P. R. China. [4]These authors contributed equally: Yong Sheng, Hengyu Wang. ✉e-mail: zxdeng@sjtu.edu.cn; bailq@sjtu.edu.cn; qjkang@sjtu.edu.cn

CRISPR-Cas defense systems are a double-edged sword since they can potentially damage beneficial foreign DNA while protecting the host, and occasionally, abrogation of CRISPR-Cas systems by MGEs has been observed. For example, prophage can integrate into CRISPR array sequences[24], and IS insertion into CRISPR loci has been observed under certain environmental conditions[25], generating susceptibility to genetic predation.

In this work, our initial prospecting reveals diverse occurrences of IS insertions into *cas* genes, likely resulting in abrogation of CRISPR immunity systems in many prokaryotes. We hypothesize that the collapse of CRISPR machinery could increase host's susceptibility to beneficial foreign MGEs, thereby facilitating the acquisition of advantageous traits and enabling effective adaptation to envrinomental challenges. Using *Escherichia coli* as the chassis, we demonstrate an interplay between CRISPR-Cas machineries and ISs that modulates host fitness through CRISPR-Cas disruption, triggered by DNA double-strand breaks (DSBs). Remarkably, through iterative mutagenesis of IS target sites in *cas* genes, IS1 and IS10 emerge as prominent players in disrupting *cas* genes in *E. coli* DH10B, demonstrating their substantial flexibility in recognizing target sites. Furthermore, IS transposition into CRISPR-Cas is sustained during introduction of non-homologous end-joining (NHEJ) repair systems, and genome mining analysis indicates that ISs interrupt other prokaryotic defense systems. These results demonstrate key roles for ISs in countering the activity of CRISPR-Cas and other genetic defense mechanisms in prokaryotes.

## Results

### Naturally occurring transpositions of ISs into Cas-encoding genes

We first screened for ISs that naturally breached CRISPR-Cas machinery. The ISs within the ISfinder database[26] were subjected to BLASTn[27] analysis (see Methods section) using a customized CRISPRCasdb database containing the identified CRISPR-Cas gene clusters from complete genome sequences[28]. The TSD sequences flanking ISs typically represent traces of transposition events, so we conducted further manual examination to identify ISs associated with TSD sequences within the target DNA regions and discovered up to 28 IS insertions that potentially disrupted CRISPR-Cas loci in different hosts (Figs. 1 and S1). Given the diverse occurrences of ISs in CRISPR-Cas loci, we hypothesized that such "attacks" by ISs might be part of the host's fitness trade-off between genetic defense and the acquisition of potentially beneficial MGEs. To address this fitness aspect, we used BLASTn searches to determine whether any of the ISs embedded in *cas* genes were also found in the database containing numerous plasmids and bacteriophages from the NCBI Nucleotide database. Although no candidate ISs were found in bacteriophages, a few were identified in plasmids, suggesting that some IS insertions might have originated from plasmids through horizontal transmission.

Spacers derived from CRISPR arrays could provide clues about invading MGEs by matching complementary spacers to their target protospacers. Therefore, we searched the plasmid database for spacer sequences from the CRISPR arrays of the 28 different CRISPR-Cas loci and identified multiple sites within 386 different plasmids that could potentially be targeted by CRISPR-Cas systems in five of the strains harboring ISs in their Cas regions (Supplementary Data 1). The CRISPR-Cas disruptions could therefore render the strains susceptible to invasion by these plasmids or other MGEs with these sequences.

However, considering that the ISs in the ISfinder repository encompass only a fraction of the ISs present in the public databases, it is conceivable that our analysis based on the ISfinder database, has only revealed a portion of the actual transposition events involving the insertion of ISs into *cas* genes. To overcome the inherent limitations of the database, we further employed a highly sensitive software pipeline, ISEScan[29], which was based on profile hidden Markov models (HMMs) constructed from manually curated ISs, thereby enabling a more comprehensive detection of ISs transpositions. As expected, we identified a total of 163 natural ISs transpositions into *cas* genes within the same CRISPRCasdb database (Fig. S2), involving the participation of 20 distinct IS families (Fig. S3) and spanning across 57 different genera (Supplementary Data 2). Additionally, we also systematically explored the transposition events of the miniature inverted repeat transposable elements (MITEs)[30], which are the non-autonomous IS derivates and can be potentially catalyzed in trans by the transposase of a related complete IS, within the *cas* genes. We then analyzed the presence of MITEs within *cas* genes of the CRISPRCasdb database using MITE-Tracker pipeline[31], revealing a similar transposition phenomenon observed with ISs (Fig. S4). In conclusion, these findings suggest that the transpositions of ISs into *cas* genes are likely not random events but rather outcomes of microbial evolutionary processes.

### IS deactivation of CRISPR-Cas facilitates rapid acquisition and dissemination of MGEs

Our analysis revealed that the phytopathogen *Erwinia amylovora* 7-3 (GenBank accession no. CP063697.1) contains an IS10-disrupted type I-E CRISPR-Cas system with specific spacers that could potentially target 237 plasmids (Fig. 1 and Supplementary Data 1). Through bioinformatic analyses of these plasmids, we noted that IS10 sequence was found only in plasmid p7-3 (GenBank accession no. CP063698.1), which is present in *E. amylovora* 7-3. Within p7-3, we identified the type IV secretion system, which may enable conjugative transfer, and streptomycin resistance genes (*strA* and *strB*), which potentially confer a host survival advantage. However, *E. amylovora* 7-3 harbors two CRISPR arrays with 11 spacers that potentially target p7-3, suggesting that a functional CRISPR-Cas system would eliminate this plasmid. We hypothesized that p7-3 evaded host immunity by IS10 insertion into the *cas* genes and that an intact CRISPR-Cas system in *E. amylovora* 7-3 would negatively affect p7-3 transfer and stability, unless IS10 again disrupted the *cas* genes.

We heterologously expressed the *E. amylovora* 7-3 type I-E CRISPR-Cas system, using plasmid pEraCas, in the surrogate *Escherichia coli* DH10B host SYHY01 (Fig. 2a) to perform CRISPR interference assays. We also constructed plasmid p15A-Eraprotos-spe, which confers chloramphenicol and spectinomycin resistance, is compatible with pEraCas, and contains all of the engineered protospacers from p7-3 and the protospacer adjacent motif (PAM) sequence of type I-E CRISPR-Cas. Compared to the control plasmid p15A-spe lacking protospacers, the transformation efficiency of p15A-Eraprotos-spe into SYHY01 decreased significantly, by ~30-fold. However, in strain SYHY02, which harbors an exogenous *E. amylovora* type I-E IS10-aborted CRISPR-Cas system (Fig. 2a), p15A-Eraprotos-spe transformation efficiency was completely restored, indicating loss of interference (Fig. 2b). Notably, many transformants emerged under selective conditions and could transiently tolerate the active CRISPR/Cas system of SYHY01 without any detectable mutations. Nevertheless, the reduced growth of SYHY01::p15A-Eraprotos-spe under antibiotic selection suggested a strong fitness cost (growth defects), and this reduction was negated in SYHY02::p15A-Eraprotos-spe (Fig. 2c). In a plasmid stability assay, the percentage of chloramphenicol/spectinomycin double-resistant clones in the CRISPR interference-positive strain (SYHY01::p15A-Eraprotos-spe) dropped dramatically to almost zero after just one passage in LB broth without antibiotics (chloramphenicol and spectinomycin), while over 25% of the interference-negative strain (SYHY01::p15A-spe, SYHY02::p15A-spe, and SYHY02::p15A-Eraprotos-spe) population still stably retained the resistance plasmid after 3 passages (Fig. 3a). These results supported the hypothesis that IS10 acted as an "off-switch" for CRISPR-Cas, facilitating maintenance of an antibiotic resistance plasmid containing identifiable protospacers, while conferring a host survival advantage under antibiotic stress.

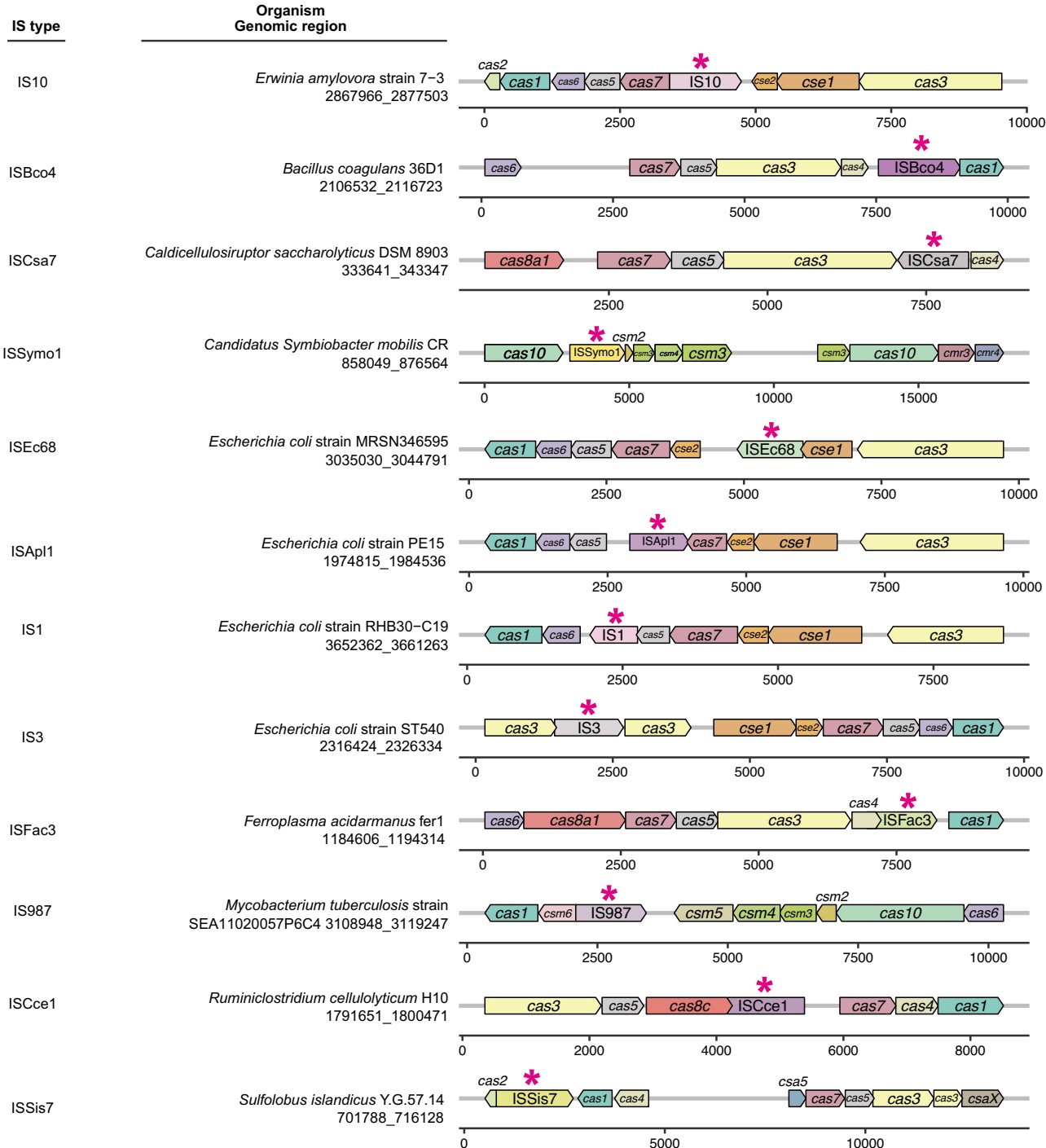

**Fig. 1 | Naturally occurring transpositions of IS elements into CRISPR-Cas loci.** Representative examples of each IS element found in CRISPR-Cas loci are shown. Arrows represent ISs (marked by asterisks) and genes within the CRISPR-Cas clusters, with arrow orientation indicating the direction of IS insertion or gene transcription. Genomic coordinates beneath the species names indicate the range of each *cas* cluster, with the numbered lines indicating the relative position of each gene or IS within the corresponding region. All 28 of the identified naturally occurring transposition events are listed in Fig. S1.

## ISs mediate the fitness trade-off between genetic defense and benefits of plasmid acquisition

As plasmid replication may proceed more rapidly than CRISPR/Cas cleavage, some plasmids with matched protospacers could be transiently maintained despite active CRISPR/Cas systems (Fig. 2b). Our plasmid stability assays (Fig. 3a) also indicated that the cumulative effect of CRISPR interference was indeed necessary for complete elimination of the target plasmid. To exclude the possibility of CRISPR/Cas interference complexes competing with continuous plasmid replication, we inserted all protospacers designed from p7-3 into the chromosome of *E. coli* DH10B to generate strain SYHY03, and then utilized pEraCas for co-expression of the type I-E *cas* operon with guide RNA to target the SYHY03 chromosome, which would be detrimental to bacterial growth. Such genotoxic stress may select for mutations that alleviate the conflict between CRISPR lethality and stable maintenance of CRISPR plasmids encoding antibiotic resistance genes. Unexpectedly, although there was a slight reduction in transformation efficiency of pEraCas into SYHY03 compared to that of pEraCas-IS10

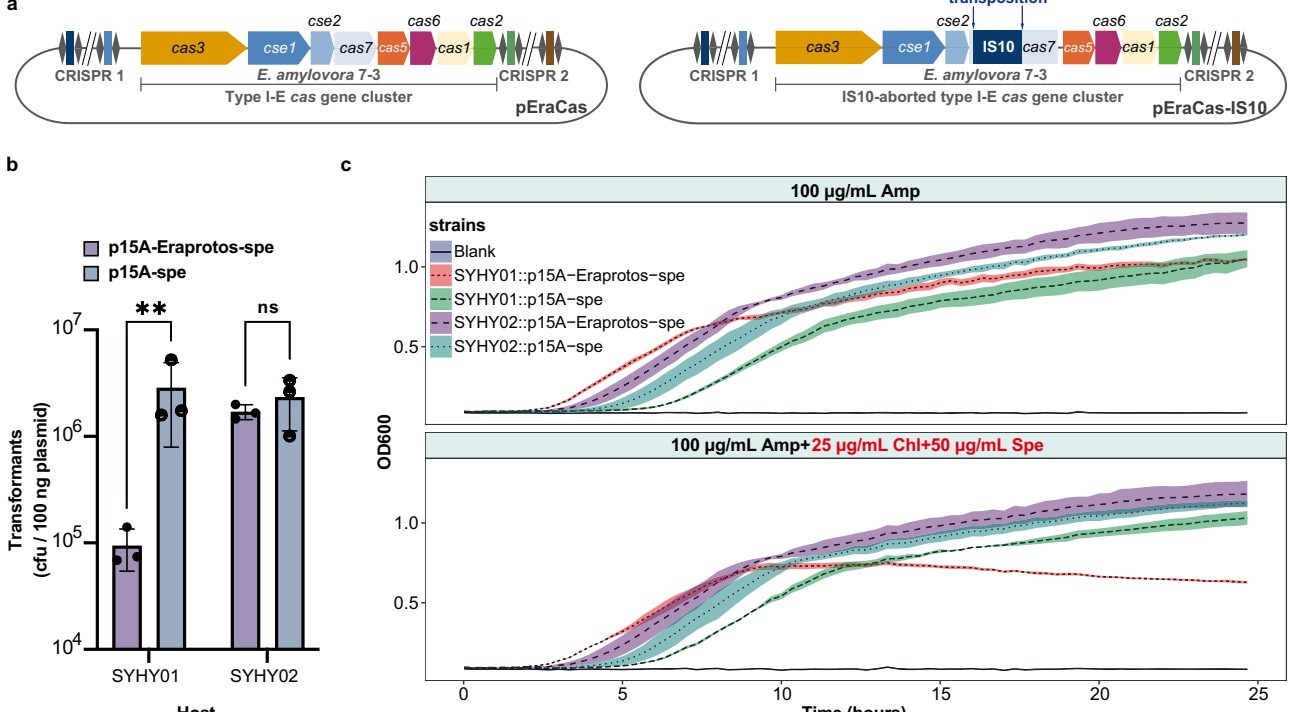

**Fig. 2 | ISs-deactivated CRISPR-Cas system facilitates the rapid acquisition and improved persistence of the antibiotic resistance plasmid containing identifiable protospacers. a** Schematic of the engineered type I-E CRISPR-Cas locus of *E. amylovora* 7-3. An intact or IS10-aborted CRISPR-Cas locus was cloned into plasmid pEraCas or pEraCas-IS10, respectively, for CRISPR-Cas interference assays. **b** Plasmid transformation interference by the *E. amylovora* 7-3 type I-E CRISPR-Cas system in *E. coli* DH10B cells. Transformation efficiency was determined by counting colony-forming units (CFUs) per 100 ng of plasmid. Data shown as means ± S.D. from three biological replicates; **$P < 0.01$; ns not significant, using two-tailed unpaired t-test of log10 transformed data; $P = 0.0018$, 0.6150 (from left to right). Source data are provided as a Source Data file. **c** Growth curves of test strains under different antibiotic exposures. Shaded areas indicate the means ± S.D. from three biological replicates. Ampicillin (Amp) was used to maintain pEraCas or pEraCas-IS10 in the host, and chloramphenicol (Chl) and spectinomycin (Spe) were used to maintain p15A-Eraprotos-spe and p15A-spe. Source data are provided as a Source Data file.

which contained an IS10-aborted CRISPR-Cas system, a large number of transformants were still observed in CRISPR interference assays, instead of inducing a canonical DNA damage (Fig. 3c). This may be due to a low level of *cas* operon transcription in the heterologous host, we therefore generated pEraCas-pBAD, in which *cas* operon expression is driven from the L-arabinose-inducible pBAD promoter. Reverse transcription quantitative PCR (RT-qPCR) analysis confirmed significant upregulation of all eight *cas* operon genes of pEraCas-pBAD after L-arabinose induction, with *cse1* and *cse2* upregulated over 95-fold (Fig. 3b). Additionally, in the absence of L-arabinose, pEraCas-pBAD transformation efficiency was comparable to that of pEraCas, whereas transformation efficiency was reduced by three logs when *cas* operon was significantly upregulated (Fig. 3c), indicating over-expression of the *cas* operon induced genotoxic stress. To investigate mutations that enabled host escape from CRISPR-Cas cleavage, we amplified and sequenced the whole *cas* operon region (Fig. 3d, e), and identified that some compensatory mutations corresponded to IS1 or IS10 transposition (Fig. 3f).

Moreover, we also utilized the endogenous type I-E CRISPR/Cas system of *E. coli* DH10B to further explore the role of ISs in balancing genetic defense and the benefits of plasmid acquisition. *E. coli* DH10B is equipped with a type I-E CRISPR/Cas system comprising eight *cas* genes (*cas1*, *cas2*, *cas3*, and *casABCDE*) and two CRISPR arrays (CRISPR I and CRISPR II)[32]. These eight *cas* genes were reported to be organized in two operons (*casA-casB-casC-casD-casE-cas1-cas2* and *cas3*). Under normal conditions, the type I-E CRISPR/Cas system of *E. coli* DH10B remains silent due to strong repression of *casABCDE12* operon by the histone-like nucleoid structuring protein (H-NS)[33,34]. In light of this, we replaced the native promoters of *casABCDE12* operon and *cas3* with a

constitutive promoter J23119 and an arabinose-inducible promoter (pBAD), respectively, generating strain ActSY01 (Fig. S5a). Additionally, we designed and inserted two different CRISPR spacers, specifically targeting the protospacers immediately downstream of the AAG (PAM) sequence within both an essential gene (*ftsA*) and a non-essential gene (*arpA*), into a p15A-derived plasmid, generating p15A-CRISPR::ftsA and p15A-CRISPR::arpA plasmids, respectively. Compared to the control plasmid p15A-sgRNA::lacZ02 lacking the targeting spacer for the type I-E CRISPR/Cas system, the transformation efficiencies of p15A-CRISPR::ftsA and p15A-CRISPR::arpA into the strain ActSY01 were dramatically decreased by about four orders of magnitude in the presence of L-arabinose induction (Fig. S5b). Nevertheless, a large number of escapers, capable of evading CRISPR/Cas-mediated cleavage, emerged under the selective pressure of chloramphenicol antibiotic. To further explore the underlying immune escape mechanisms, we randomly selected 23 escapers from the strains ActSY01 harboring p15A-CRISPR::ftsA or p15A-CRISPR::arpA, and subsequently performed PCR amplification and Sanger sequencing of the complete chromosomal *cas* operons. As expected, we detected various compensatory mutations corresponding to the transpositions of IS1, IS2, IS5, and IS10 into the *cas* genes within the intrinsic type I-E CRISPR/Cas system of the strain ActSY01 (Fig. S5c and Table S4).

These results showed that IS insertions into the *cas* operon could alleviate the conflict between CRISPR/Cas-induced genotoxicity and target maintenance. Given these observations and our bioinformatic analyses, we hypothesized that ISs can be harnessed in the fitness trade-off between the potential benefits of acquiring foreign DNA under environmental stress and the need for genetic defense.

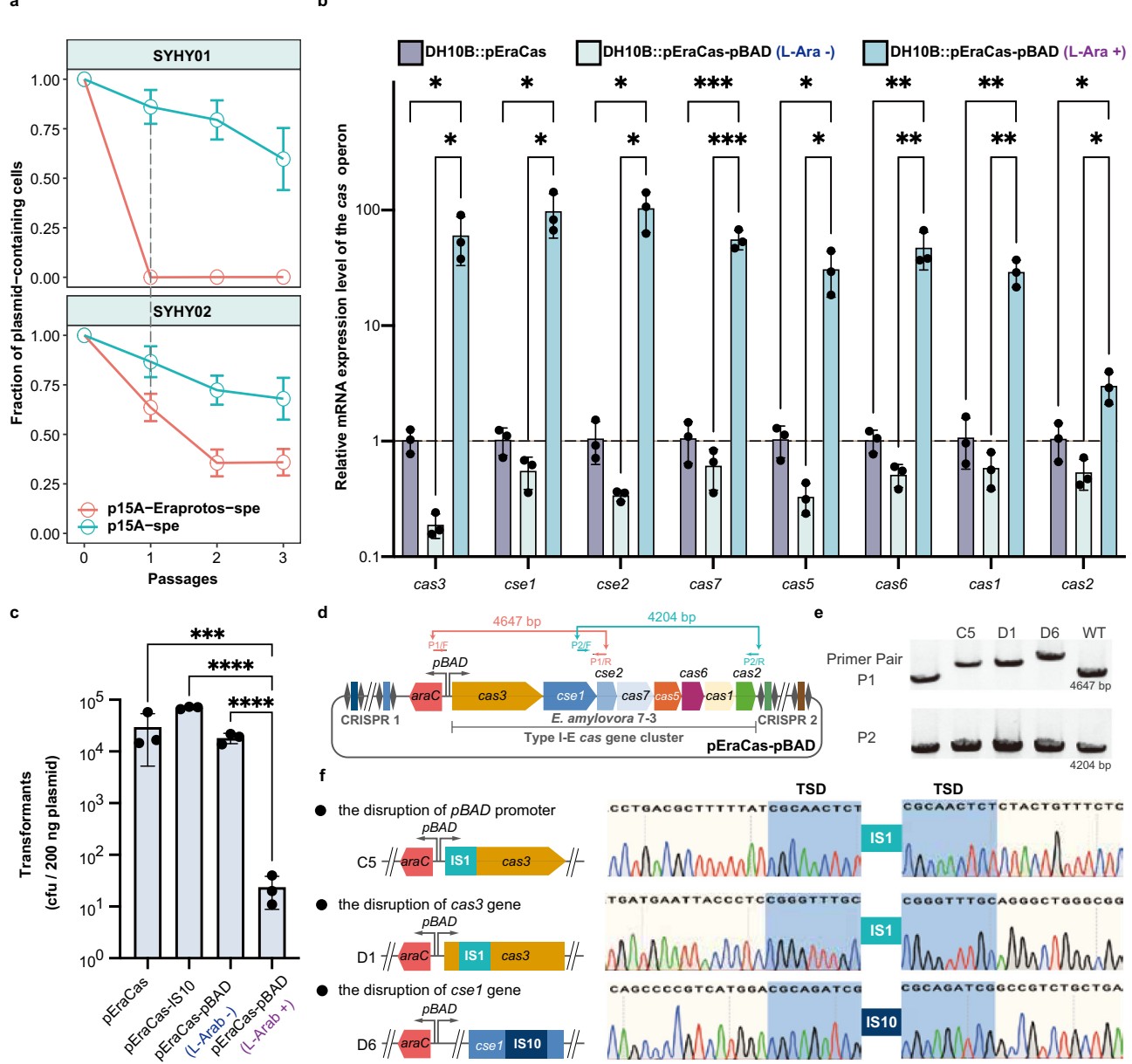

**Fig. 3 | ISs mediate the fitness trade-off between the benefits of acquiring plasmids and genetic defense under selective pressure. a** Plasmid stability after transformation and during passage in SYHY01 and SYHY02. Values shown as mean ± SEM from three biological replicates. Source data are provided as a Source Data file. **b** Relative transcription levels of eight genes within the *cas* operon of pEraCas and pEraCas-pBAD in *E. coli* DH10B. Gene expression was quantified by RT-qPCR and normalized to levels of endogenous 16sRNA. Error bars denote means ± S.D. from three biological replicates; *$P < 0.05$; **$P < 0.01$; ***$P < 0.001$, using two-tailed unpaired *t*-test; all the presented *P*-values were displayed from top to bottom; 0.019, 0.019 (*cas3*); 0.015, 0.014 (*cse1*); 0.011, 0.011 (*cse2*); 0.000886, 0.000858 (*cas7*); 0.017, 0.016 (*cas5*); 0.009, 0.009 (*cas6*); 0.003, 0.003 (*cas1*); 0.028, 0.011 (*cas2*). Source data are provided as a Source Data file. **c** Transformation efficiencies of pEraCas (targeting the SYHY03 chromosome), pEraCas-IS10 (control), and pEraCas-pBAD (targeting the SYHY03 chromosome in the presence of L-

arabinose) into SYHY03 were determined through counting CFUs per 200 ng of plasmids. Error bars denote means ± S.D. from three biological replicates; ***$P < 0.001$; ****$P < 0.0001$, using two-tailed unpaired *t*-test of log10 transformed data. $P = 0.000256, 2.66e{-}05, 7e{-}05$ (from top to bottom). Source data are provided as a Source Data file. **d** Schematic of pEraCas-pBAD harboring the L-arabinose-inducible pBAD promoter to drive *cas* operon expression. Primer pairs P1 and P2 were designed to amplify the entire *cas* operon of pEraCas-pBAD; expected amplicon sizes are shown. **e** Colony PCR screening with primer pairs P1 and P2 for IS insertions into the pEraCas-pBAD *cas* operon. Lanes "WT", control amplicons with the pEraCas-pBAD template; larger PCR products (lanes C5, D1, and D6) suggest IS transposition events. The experiment was conducted independently three times and yielded consistent results. Source data are provided as a Source Data file. **f** Confirmation by Sanger sequencing of IS insertions into *cas* operons of mutants C5, D1, and D6. Shaded areas of chromatograms, TSDs resulting from IS insertions.

## An IS trapping system to explore the IS-mediated fitness trade-off

We developed a CRISPR/Cas-mediated IS trapping system that harbored a genetic circuit for monitoring specific IS attacks and that was based on the inhibition of Cas-induced DSBs by IS insertion into the *cas* gene (Fig. 4a). Our search for ISs within the genomes of *E. coli* MG1655

and *E. coli* DH10B identified eight diverse IS families with varied copy number, which offered the desired chassis for investigations on the complexity and diversity of ISs-mediated fitness trade-off (Fig. S6). We developed plasmid interference trials using the type II (SpCas9) CRISPR/Cas system with a single multidomain effector protein arbitrating both recognition and cleavage to monitor IS insertions into the

**a**

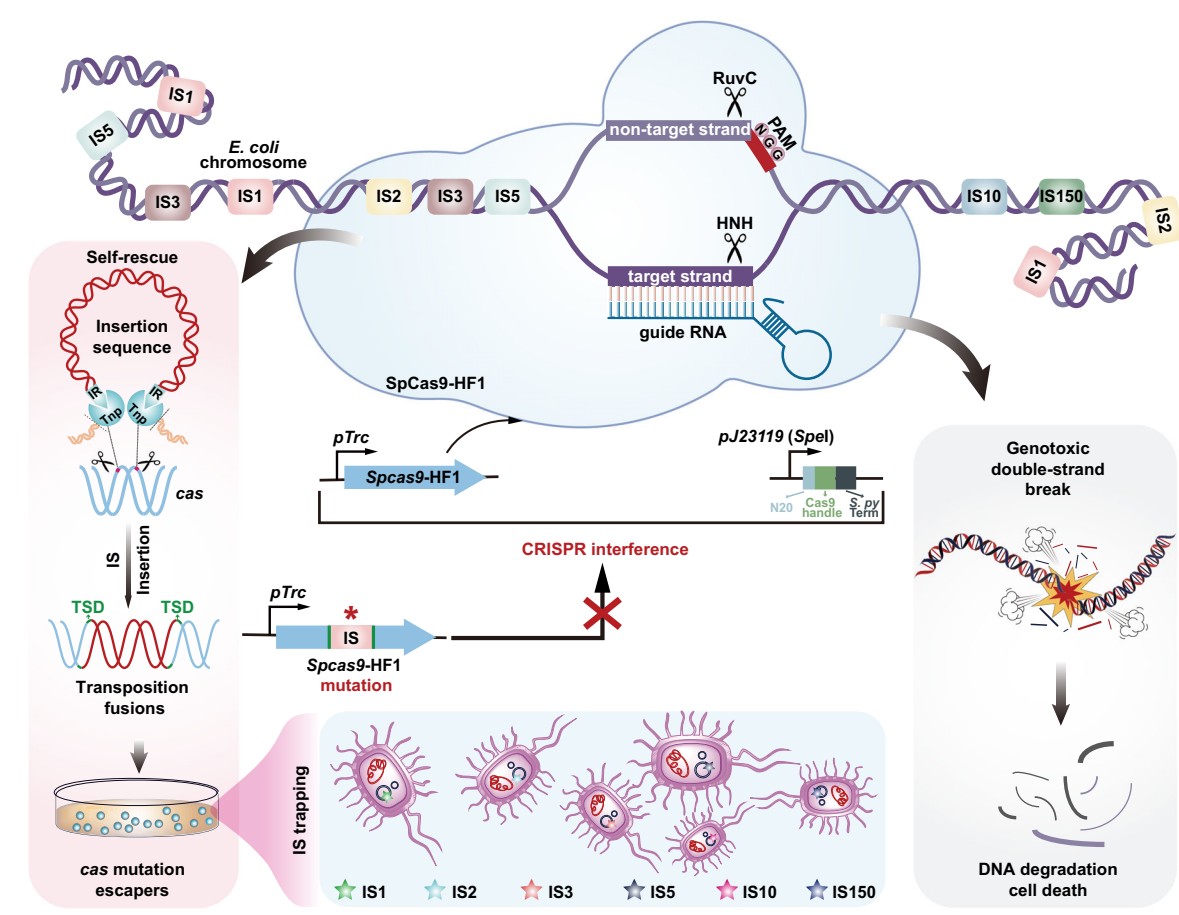

**b**

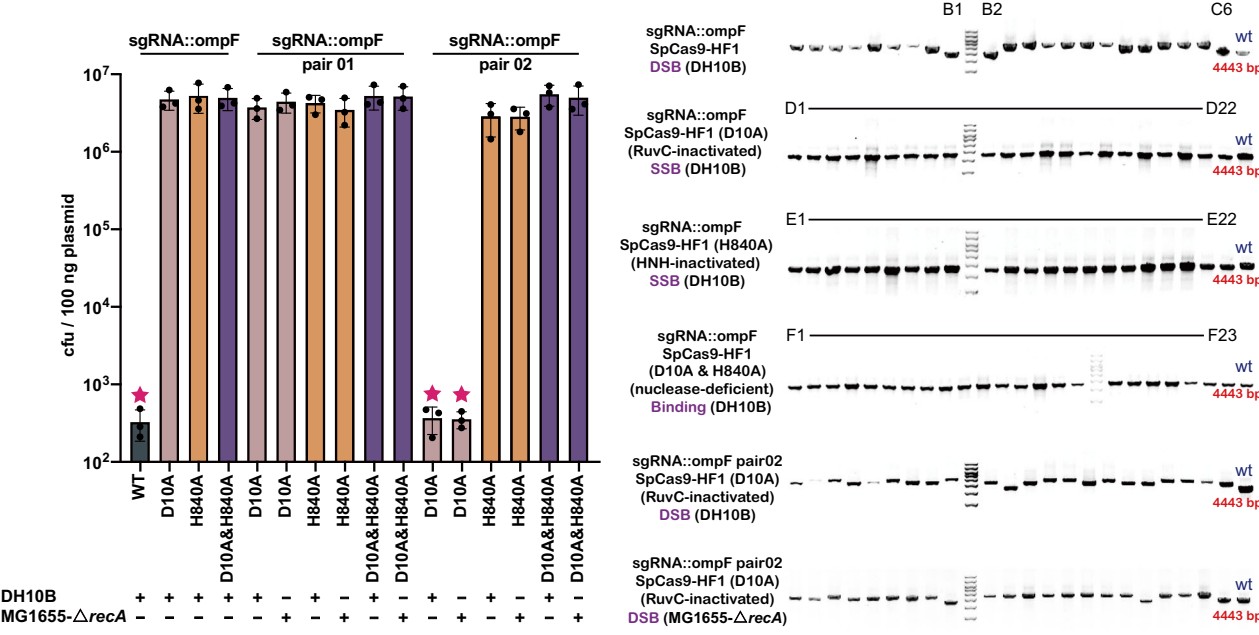

**c**

*cas* gene in the two *E. coli* hosts (Fig. 4a). Such surrogate *E. coli*-based artificial CRISPR interference system presented a notable advantage over the complex type I-E CRISPR/Cas system, as it only required the detection of a single gene during ISs transpositions analysis, while the latter necessitated the simultaneous identification of eight genes. SpCas9-HF1 is a high-fidelity SpCas9 variant that displays higher specificity and lower off-target effects than the original system[35]. In our IS-

trapping system, expression of the SpCas9-HF1 nuclease was driven by the IPTG-inducible promoter pTrc (Fig. 4a). Recipient strains were first transformed with the plasmid containing the SpCas9-HF1-expressing cassette and then with a plasmid expressing single guide RNA (sgRNA) that targets the *ompF* gene within the host genomes. Following incubation under antibiotic selection pressure, induction of SpCas9-HF1 expression by IPTG would result in potentially lethal DSBs in the *ompF*

**Fig. 4 | Establishment and characterization of the CRISPR/Cas-mediated IS trapping system. a** Schematic of the IS-trapping system in which DSBs trigger transposition of IS elements, including insertions into *cas*. **b** Transformation efficiency of the sgRNA-expressing interference plasmid in *E. coli* DH10B and MG1655-Δ*recA* harboring CRISPR-Cas (WT) or CRISPR-Cas mutants. Mutants were generated by site-directed mutagenesis in D10A and H840A within the SpCas9-HF1 coding region, inactivating the RuvC and HNH domains, respectively. sgRNA::ompF, sgRNA::ompF pair 01, and sgRNA::ompF pair 02 represent targeting plasmids p15A-cm-sgRNA::ompF, p15A-cm-sgRNA::ompF pair 01, and p15A-cm-sgRNA::ompF pair 02, respectively. The five-pointed star above bars indicates the strain has a CRISPR/Cas-induced double-strand break in the chromosome. The "+" or "−" indicate that the given *E. coli* strain was used or not used in the corresponding sample in the bar graph. WT, D10A, H840A, and D10A&H840A indicate plasmids pCasY-3-ΔSIM-kan, pCasY-3-ΔSIM-kan (D10A), pCasY-3-ΔSIM-kan (H840A), and pCasY-3-ΔSIM-kan (D10A&H840A), respectively; error bars indicate means ± S.D. from three biological replicates. Source data are provided as a Source Data file. **c** PCR screening for IS insertions into *cas* gene expression cassettes. The single colonies were taken from each of the four types of CRISPR-tolerant mutants from Fig. 4b, simultaneously containing the genomic target sequence, sgRNA expression plasmid, and CRISPR-Cas expression cassette [pTrc-SpCas9-HF1, pTrc-SpCas9-HF1 (D10A), pTrc-SpCas9-HF1 (H840A), or pTrc-SpCas9-HF1 (D10A&H840A)], followed by colony PCR validation. The expected size of PCR amplicons from the four types of *cas* gene expression cassettes without IS element insertions was 4443 bp; larger PCR products indicate IS insertion. DSB denotes double-stranded DNA breaks, while SSB represents single-stranded DNA breaks. Binding refers to the specific binding of Cas protein to the target site without inducing DNA cleavage. The experiment was conducted independently three times and yielded consistent results. Source data are provided as a Source Data file.

gene targeted by the sgRNA. However, we hypothesized that through transpositions of ISs into *cas* genes, hosts could achieve coexistence with plasmids that targeted the chromosome, effectively eliminating the CRISPR/Cas-induced genotoxicity while simultaneously acquiring antibiotic resistance gene encoded by the plasmids, thereby enabling survival under antibiotic pressure.

Generally, classical IS elements can transpose autonomously and generate short TSDs in target DNA. Such transposition fusions (Fig. S7) can be detected by PCR and confirmed by Sanger sequencing. RecA activity was another consideration in our system. RecA can mediate genomic DNA damage repair through recombination between distant homologous sequences or between microhomologies[36], which may lead to emergence of survivor colonies unrelated to IS insertions. Therefore, for CRISPR interference studies, we generated *E. coli* MG1655-Δ*recA*, a *recA* disruption mutant of *E. coli* MG1655, and also utilized *E. coli* DH10B, a recombination-defective (*recA1*) strain. IS-free strain *E. coli* MDS42-Δ*recA* was used as control.

Survivor cells (colonies) in the transformation assays were counted and analyzed. Although the *ompF*-targeting sgRNA plasmid exhibited low electrotransformation efficiency, we still discovered colonies using *E. coli* MG1655-Δ*recA*, DH10B, and MDS42-Δ*recA*. To explore the mechanisms underlying the appearance of the "escaper" (survivor) colonies, the genetic components essential for proper functioning of the CRISPR-Cas system were analyzed by PCR and DNA sequencing. Remarkably, most PCR amplicons of the SpCas9-HF1 expression cassette were larger in MG1655-Δ*recA* and DH10B than in MDS42-Δ*recA* survivors, which had amplicons of the expected size of 4443 bp. Sequencing of 50 amplicons, including 44 of the larger PCR products and 6 fragments of the expected size, was carried out and revealed that the larger amplicons had an IS element inserted into the SpCas9-HF1 coding region (Table S5). A very small number of escapers without IS insertions had indel mutations in the SpCas9-HF1 coding region or in the sgRNA expression cassette. These results illustrated that the CRISPR interference approach was a suitable platform for demonstrating that IS transpositions could dramatically impede CRISPR-Cas machinery.

To elucidate whether the occurrences of such IS transpositions were widespread among different targeting loci, we designed another three modular sgRNA expression cassettes to target the non-essential genes *lacZ*, *pyrF*, and *lpp* in MG1655-Δ*recA* and DH10B chromosomes to ensure that the killing by CRISPR-Cas was the result of a broken chromosome and not interference of an essential target gene. PCR surveys and DNA sequencing of SpCas9-HF1 expression cassettes for the selected survival targets revealed that the majority of insertions were due to different jumping IS elements (Fig. 5a).

As the pTrc promoter can be leaky in the absence of IPTG, we tightened control of SpCas9-HF1 levels by adding an ssrA-tag to enable rapid degradation by ClpXP protease. The SpCas9-HF1-ssrA coding region was also targeted by different IS elements in MG1655-Δ*recA* and DH10B as was FnCpf1 (Fig. 5a), a type V-A CRISPR-Cas system, which contains only the RuvC endonuclease domain without the trans-activating crRNA companion[37]. Analysis of 448 colony amplicons from our CRISPR interference study identified IS1, IS2, IS3, and IS5 in MG1655-Δ*recA* and IS1, IS2, IS3, IS5, and IS10 in DH10B (Fig. 5a), with the insertion sites randomly distributed throughout *cas* genes (Fig. 5c, d). Overall, these results indicated that various ISs could disrupt different CRISPR-Cas systems.

To investigate whether ISs transpositions into *cas* genes could potentially lead to concurrent transpositions into other genomic loci or induce genomic rearrangements and instability, we performed whole-genome sequencing of 11 CRISPR-tolerant mutants that survived after CRISPR/Cas-induced killing and an uninduced control strain, SY221, containing only the *cas*-expressing plasmid without sgRNA guides (Table S6). Sequence alignment of SY221 with the CRISPR-tolerant mutants revealed 100% sequence identity at 100% chromosome coverage, suggesting that the transpositions of ISs into *cas* genes did not result in simultaneous transpositions into other genomic loci or trigger genomic rearrangements.

## DSBs trigger IS transposition into *cas* genes

To determine whether different types of chromosomal damage influenced IS element mobility, we used site-specific mutagenesis to generate SpCas9-HF1 variants with altered DNA cleavage capacities: nickase SpCas9-HF1 (D10A) has an inactivated RuvC domain; nickase SpCas9-HF1 (H840A) has an inactivated HNH domain; and the double mutations D10A & H840A in the nuclease-deficient SpCas9-HF1 (D10A & H840A) still allowed target binding but no cleavage[38,39]. We firstly explored the impacts of the expression of SpCas9-HF1 protein with varying cleavage capabilities on the growth of bacterial strains in the absence of sgRNA targeting guides. Through monitoring the growth of the four strains encoding different Cas proteins under both IPTG-induced and non-induced conditions, we observed that the mere induction of Cas proteins did not result in discernible growth defects or impose significant physiological burdens on the host strains (Fig. S8). Subsequently, we observed higher transformation efficiencies of the plasmid harboring the sgRNA targeting guide when introduced into DH10B encoding different SpCas9-HF1 variants, compared to DH10B encoding the wild-type SpCas9-HF1(Fig. 4b). Additionally, PCR and sequencing analysis of the recovered colonies revealed that no IS insertions or other mutations impaired the proper functioning of the three CRISPR-Cas variants (Figs. 4c and S9a). These evidences suggested that IS transposition was not induced by single-strand nicks or in the absence of chromosomal nicks, with the nicks likely repaired by DNA damage repair pathways, and that these variants did not generate the genotoxic stress associated with parental SpCas9-HF1.

We then designed a double-nicking strategy to assess the correlation between IS-mediated transposable defense mechanisms and occurrence of DSBs. A pair of RNA-guided single-strand nicks anchored at opposite strands of a targeted DNA locus can generate a

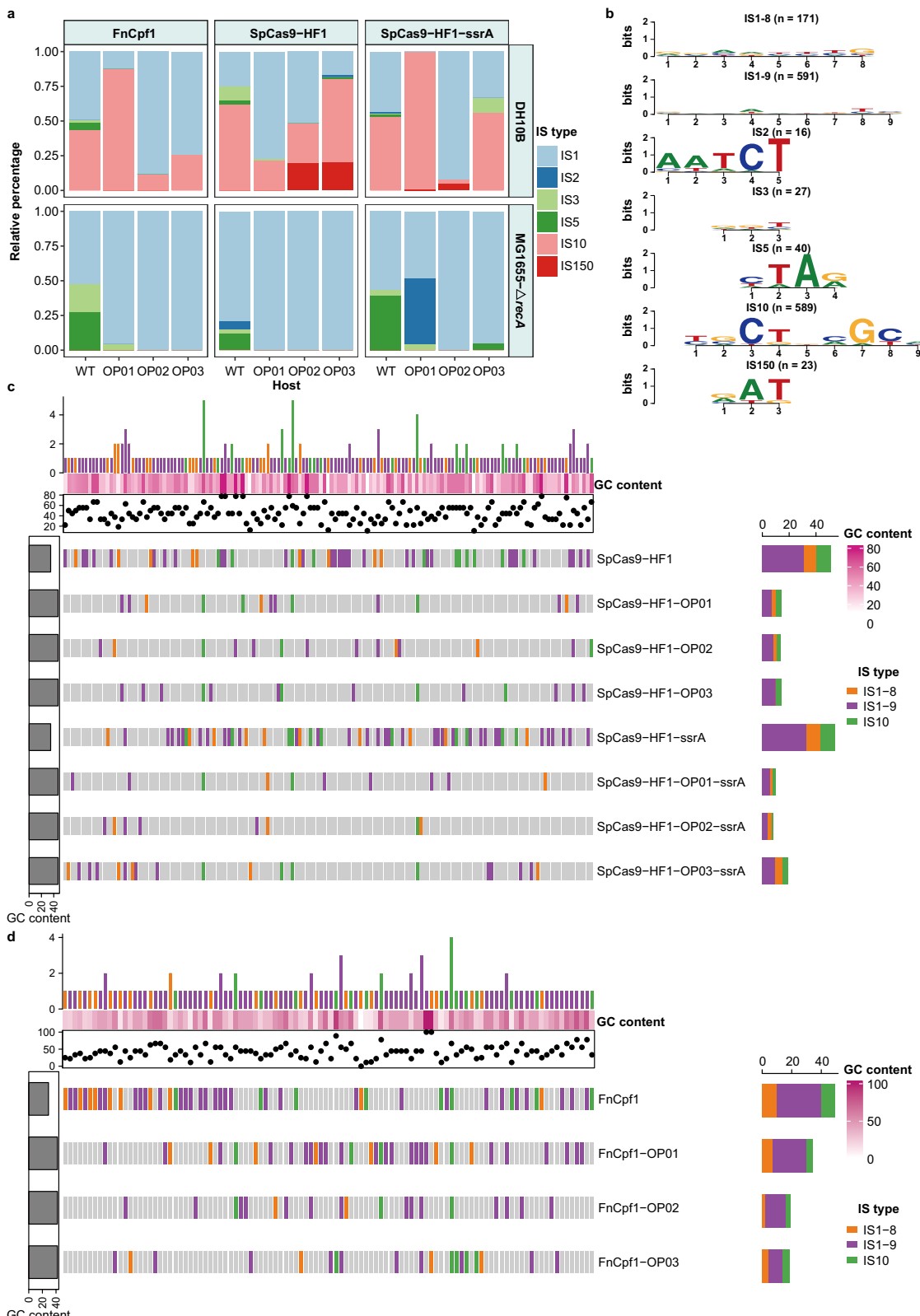

DSB, and we used two sets of sgRNA pairs targeting the *ompF* genomic locus, but with target sites offset by either +13 bp (pair 01) or +29 bp (pair 02), and evaluated their transposition potency with SpCas9-HF1 (D10A) and SpCas9-HF1 (H840A) (Fig. S9b). sgRNA pair 02 with SpCas9-HF1 (D10A) produced notably low plasmid transformation efficiency and only yielded survivor cells through IS-mediated defense, with IS insertions into SpCas9-HF1 (D10A) in both strains (Fig. 4b, c). In

contrast, sgRNA pair 01 (+13 bp offset) produced high plasmid transformation efficiency (Fig. 4b), and presumably, due to the steric hindrance of adjacent Cas molecules or Cas-sgRNA complexes, was unable to generate DSB and therefore did not initiate the IS-mediated defense response. The RuvC domain, preserved in SpCas9-HF1 (H840A) nickase, displayed lower activity than the HNH domain[40] and was insufficient to induce DSB, also resulted in high

**Fig. 5 | Fitness consequences of IS-mediated trade-off by insertion into the _cas_ gene. a** The stacked bar chart depicts the relative frequency of transpositions of each IS element (indicated by the different colors) into the coding regions of SpCas9-HF1, SpCas9-HF1-ssrA, FnCpf1, and their derivates within _E. coli_ DH10B and MG1655-Δ_recA_. Source data are provided as a Source Data file. **b** Based on Sanger sequencing results of all transposition fusions identified in **a**, consensus TSD motifs were constructed and are displayed as sequence logos for each IS type. Values within parentheses indicate the total number of TSDs detected for each IS type. **c**, **d** Comparative analysis of the diversity of IS transposition sites across the coding regions of SpCas9-HF1 (**c**), FnCpf1 (**d**), and their derivates. Vertical bars represent all unique positions of identified TSDs in the related genes, arranged from left to right in ascending order of the coordinates. Sites of TSDs generated by IS1-8 (8 bp TSD of IS1), IS1-9 (9 bp TSD of IS1), and IS10 are marked by vertical bars of orange, purple, and green, respectively. Gray vertical bars indicate that no IS insertions were observed at the site. The GC content of each TSD is represented by a dot on the scatter plots, and the color scale in the heat maps (rows of white to dark pink bars) indicates the level of GC content. The bar chart above each heat map illustrates the number of genes sharing the same TSD site, and the bar chart on the right shows the number of unique TSD sites identified in each gene. The leftmost bar represents the GC content for each gene sequence.

transformation efficiency (Fig. 4b) and absence of IS transposition into SpCas9-HF1 (H840A). Overall, these CRISPR interference analyses indicated that DSBs triggered IS mobilization into _cas_ genes.

## IS1 and IS10 dominate in counteracting the rapid and forced evolution of Cas variants

In our assays, IS1 constituted the majority of captured elements in MG1655-Δ_recA_, with IS1 and IS10 predominant in DH10B. In contrast, only some IS5 and fewer IS2 and IS3 insertions were obtained, implying significant variability in the transposition efficiency of different ISs and highlighting their differential contributions to host defense by disrupting CRISPR-Cas machinery. To generate IS-silent Cas proteins, i.e., coding sequences lacking IS target sites, we reconstructed SpCas9-HF1 and FnCpf1 using codon degeneracy to exclude as many of the discovered IS1, IS5, and IS10 insertion sites as possible (see Methods), while also eliminating duplicated sequences (≥10 bp) in the _cas_ genes to avoid homologous DNA recombination between internal sequences. After plasmid construction, the resulting SpCas9-HF1-OP01, SpCas9-HF1-OP01-ssrA, and FnCpf1-OP01 variants were evaluated for their ability to escape IS disruption (Fig. S10). For IS insertions into _cas_ in DH10B, the predominant "guard" element depended on the CRISPR-Cas system. With FnCpf1-OP01 and SpCas9-HF1-OP01-ssrA, IS10 was predominant, whereas IS1 dominated with SpCas9-HF1-OP01 (Fig. 5a). In MG1655-Δ_recA_, which lacks IS10, all of the guard ISs from the CRISPR interference escapers were IS1. Since IS5 insertion is typically accompanied by the highly conserved TSD motif YTAR, which was targeted for elimination in the reconstructed _cas_ genes, IS5 insertions were no longer detected (Fig. 5a).

The insertion hotspots identified from the reconstructed _cas_ genes were further mutated, yielding SpCas9-HF1-OP02, SpCas9-HF1-OP02-ssrA, and FnCpf1-OP02, which resulted in IS1 becoming the predominant guard element in both _E. coli_ strains, with only a relatively small number of IS10 insertions found in DH10B (Fig. 5a). We next wanted to generate _cas_ variants that could overcome transposable defense by IS1, and so the favored IS targets in SpCas9-HF1-OP02, SpCas9-HF1-OP02-ssrA, and FnCpf1-OP02 were mutated again, yielding SpCas9-HF1-OP03, SpCas9-HF1-OP03-ssrA, and FnCpf1-OP03. IS1 remained the premier defense element within MG1655-Δ_recA_. Unexpectedly, in DH10B, IS10 dominated with SpCas9-HF1-OP03, and the relative insertion ratios of IS10 with SpCas9-HF1-OP03-ssrA and FnCpf1-OP03 were further increased (Fig. 5a). Overall, through our rapid and forced evolution of _cas_ genes, anti-CRISPR defense activities in DH10B primarily alternated between IS1 and IS10, whereas IS1 dominated in MG1655-Δ_recA_.

## High variability of IS1 and IS10 TSD sequences mediates defense against _cas_ evolution

In our CRISPR interference assays, a total of 1457 IS insertions by IS1, IS2, IS3, IS5, IS10, and IS150 were obtained within the _cas_ genes (Fig. 5b). Systematic analysis of the TSD motifs revealed that IS1 had loosely conserved motifs, but showed a strong preference for insertion into 8 bp or 9 bp AT-rich regions (Fig. 6d, e) as previously established[41]. Surprisingly, our assays indicated that the recognition properties of IS1 could be greatly expanded, including the capacity to recognize 100%

GC sequences (Fig. 6e) and novel aberrant targets of 7 bp, 11 bp, and even 23 bp (Table S7). Moreover, 9 bp TSDs were present in up to 78% of the IS1 transposition events, and such attack sites were more highly dispersed across the _cas_ genes than were 8 bp TSDs, which comprised only 22% of all observations (Figs. 5c, d and 6f). Compared to IS1, IS10 exhibited greater conservation in target length and specificity; we identified 589 IS10 transposition events in which IS10 had preferentially jumped into more promiscuous hotspots (Fig. 5b) than previously established[42].

Given the loose conservation of TSD sequences for IS1 and IS10, we investigated naturally occurring transposition events induced by both ISs. We adapted a library of all previously downloaded genomic sequences of archaea, bacteria, and viruses, with a genetic mapping strategy to explore the potential relationships between IS presence and host fitness. Natural IS1 and IS10 transpositions were not detected in either archaea or viruses, whereas 1345 and 2690 transposition events for IS1 and IS10, respectively, were discovered within 19,589,090 bacterial genome sequences, mostly in _Escherichia_, _Klebsiella_, and _Salmonella_ (Fig. S11). Also, the resulting consensus TSD sequences for IS1-8, IS1-9, and IS10 were enriched from 362, 983, and 2,690 independent transposition events, respectively (Fig. 6d). Notably, the enriched TSD motifs were largely consistent with the profiles established in our CRISPR interference assays, with IS1 exhibiting a preference for 9 bp TSDs (Fig. 6f).

To determine whether IS1 and IS10 TSD motifs were conserved throughout our forced evolution of _cas_, the consensus motifs from each evolutionary round (Fig. S12) were placed into a position frequency matrix and subsequently assessed by the Pearson correlation coefficient. Generally, TSD motifs from unmodified _cas_ correlated most strongly with naturally occurring TSDs, whereas correlations decreased to varying degrees during the reengineering process (Fig. 6a–c), strongly suggesting that IS1 and IS10 could disrupt _cas_ genes at non-canonical and canonical targets and that targeting of non-canonical motifs would increase with continuous deliberate mutation of target sequences.

Further comparison revealed that IS1 TSD motifs exhibited lower sequence conservation during each evolutionary cycle, with IS10 motifs more conserved with preferred base pairs at positions 3, 4, 7, and 8 (Figs. 6d and S12). Remarkably, the TSD features identified in our CRISPR/Cas-mediated IS-trapping system were highly consistent with those occurring naturally, indicating CRISPR interference was a very reliable system to investigate IS mobility. However, the emergence of non-canonical motifs indicated that IS1 and IS10 possessed significantly broader target spectrums than originally believed, which could enhance the potential for these ISs to mediate the balance between genetic defense and evolution with regard to CRISPR interference.

## IS transpositions in presence of NHEJ repair machinery

DSBs induced by CRISPR-Cas interference are generally lethal. However, the prokaryotic non-homologous end joining (NHEJ), a DSB repair mechanism, is initiated by Ku protein binding to the DSB ends, which subsequently recruits DNA ligase D (LigD) to process incompatible DNA termini for further ligation[43]. We explored IS-mediated

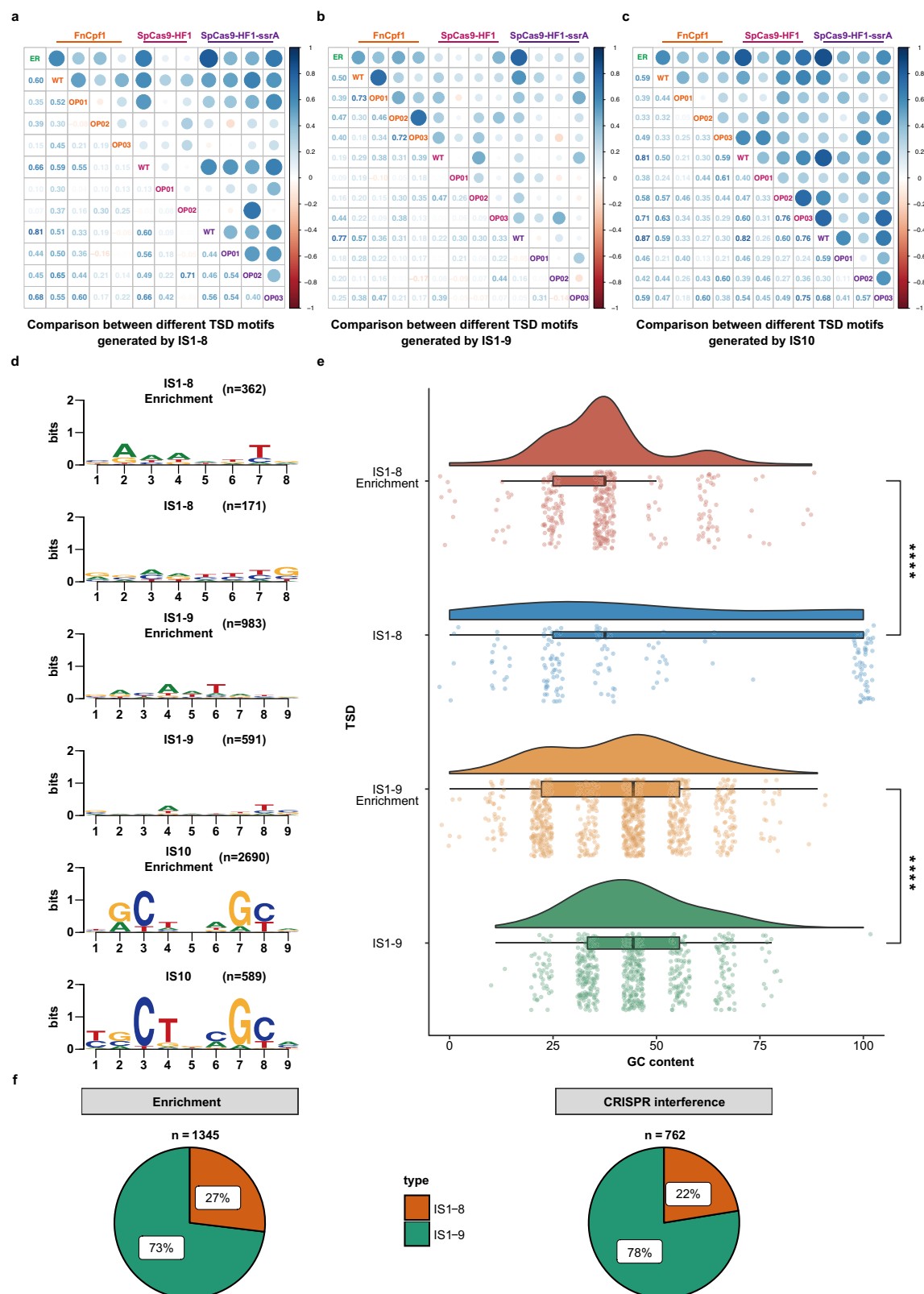

defense against CRISPR-Cas interruption following the introduction of an NHEJ system in *E. coli*, using three NHEJ cassettes with different repair capabilities: *Bacillus subtilis* 168 (BsNHEJ), *Mycobacterium smegmatis* mc²155 (MsNHEJ), and *Mycobacterium marinum* M (MmNHEJ). For each NHEJ system, we synthesized and cloned the two-component DNA repair cassettes into a ColE1-origin plasmid with the *ku* gene under the control of the pBAD promoter (Fig. 7a). Using PCR

validation, IS transposition into SpCas9-HF1 was assayed in MG1655-Δ*recA* cells in the presence and absence of induction of NHEJ and using the *ompF* gene as the CRISPR-Cas target.

During NHEJ-mediated DSB repair, indel mutations are frequently introduced at the repaired sites. Accordingly, the targeted *ompF* site from 81 randomly selected mutants without IS transposition into SpCas9-HF1 coding region was sequenced to evaluate the repair

**Fig. 6 | Comparative analysis of the TSD motifs resulting from transpositions of IS1 and IS10 into different target loci.** Pearson correlation analysis for all TSD motifs of IS1-8 (**a**), IS1-9 (**b**), and IS10 (**c**) transpositions. The TSD motifs of the naturally occurring IS1 and IS10 transpositions were gathered by screening the RefSeq database and marked with the abbreviation "ER" in each category. Pearson's correlation coefficients concerning both ER and the TSD motifs obtained through our established CRISPR/Cas-mediated ISs trapping system were calculated and displayed on a color scale. The values in the left lower triangle indicate the Pearson coefficients. Generally, the bluer the color, the stronger the correlation coefficient, suggesting the more similar composition of TSD motifs. **d** Naturally occurring TSD motifs (Enrichment) together with the experimentally determined TSD motifs of IS1-8, IS1-9, and IS10 are presented. The values in parentheses indicate the number of identified transposition events. **e** Analysis of GC content distribution for each TSD sequence of IS1-8 and IS1-9 derived from the CRISPR/Cas-mediated ISs

trapping system and naturally occurring transpositions. The values corresponding to each data set are shown in the jitter scatter plot at the bottom panel, and the statistical significance of the differences between each compared group was determined using two-tailed unpaired *t*-test; the datasets IS1-8 Enrichment, IS1-8, IS1-9 Enrichment, and IS1-9 corresponded to a total of 362, 171, 983, and 591 biologically independent samples, respectively; ****$P < 0.0001$; $P = 1.3e-06$, $5.6e-06$ (from top to bottom). The boxplots show medians (midlines), interquartile ranges (boxes), and ranges (whiskers). Source data are provided as a Source Data file. **f** Comparison of the relative ratio of different TSD motif lengths of IS1 identified from naturally occurring transposition events (Enrichment) and the CRISPR/Cas-mediated ISs trapping system (CRISPR interference). Noticeably, IS1-9, which generated a 9 bp TSD sequence, dominated in both datasets, indicating that IS1 was more likely to produce 9 bp TSD sequences.

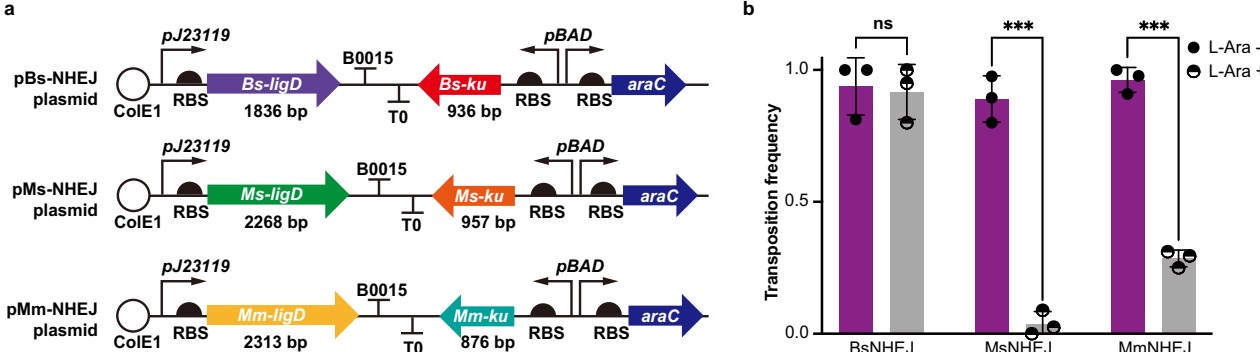

**Fig. 7 | IS transposition into *cas* genes in the presence of NHEJ. a** Schematic diagram showing the assembly of the three NHEJ systems, which include the *ku* and *ligD* genes involved in NHEJ repair. **b** Transposition frequency was assayed in MG1655-Δ*recA* harboring *ompF*, the target gene for DSBs mediated by SpCas9-HF1/sgRNA, followed by error-prone NHEJ DNA repair. Addition of L-arabinose (10 mM) induced *ku* expression. Up to 45 surviving colonies per plate were randomly

selected for PCR validation, and the proportion of IS insertions into the SpCas9-HF1 coding region was calculated for each condition. Graph shows means ± S.D. derived from three biological replicates. ns, not significant; ***$P < 0.001$, using two-tailed unpaired *t*-test; $P = 0.822$, $0.000119$, $3.3e-05$ (from left to right). Source data are provided as a Source Data file.

capability of the three NHEJ systems (Fig. S13). MsNHEJ with 21/29 survivors containing indel mutations was superior to that of the MmNHEJ system with 10/35 survivors carrying indel mutations, whereas BsNHEJ displayed inferior repair ability with few rescued clones among the escapers. Notably, IS transpositions into SpCas9-HF1 coding region still occurred despite NHEJ repair activity, indicating that NHEJ was not sufficient to completely repair the CRISPR/Cas-induced chromosomal DSBs. Transposition frequency analysis revealed a negative correlation between NHEJ repair ability and the number of IS transpositions into SpCas9-HF1 coding region (Fig. 7b), the majority of which were IS1. These observations revealed that ISs provide a protective function even in the presence of NHEJ repair and strongly suggested that IS attacks play a critical role in safeguarding bacterial populations from genetic alterations.

### ISs interact with diverse prokaryotic genetic defense systems

We next analyzed the distribution of IS families in prokaryotic genomes. A collection of the known IS elements were subjected to BLASTn analysis against the full NCBI RefSeq database of unique genomic sequences of archaea, bacteria, and viruses, followed by taxonomic grouping of the genomes. IS families exhibited patchy taxonomic distribution among bacteria (Fig. S14) and archaea (Fig. S15), and copy numbers of various IS families varied greatly from 1 to 103 copies per cell (Figs. S16 and S17), whereas IS families were poorly represented in viruses (Figs. S18 and S19). The most frequent and abundant IS families in archaea and bacteria were distributed in the genera *Methanosarcina* and *Halobacterium* and class Gammaproteobacteria, with the Gammaproteobacteria genera *Escherichia*, *Klebsiella*, and *Salmonella* having the richest diversity (Figs. S20 and S21). Such wide distribution

strongly implied that IS elements play a crucial role in the physiological maintenance or evolutionary drive of the hosts.

To better understand the role of IS elements in host evolution, we explored their interactions with other prokaryotic defense systems. Based on Hidden Markov Model profiles, we searched for genes corresponding to 60 different anti-phage systems within the 10 kb regions upstream and downstream from the target IS (IS1 and IS10) landing pads. Intriguingly, we discovered insertions of IS1 and IS10 into 12 different defense gene clusters, such as RM and Dnd, in addition to the CRISPR-Cas system (Figs. 8 and S22). Such IS insertions potentially disabled the targeted immunity machineries and likely increased the susceptibility of these hosts to MGEs.

## Discussion

ISs have long been considered as selfish or parasitic MGEs since they only encode transposases necessary for their own proliferation[44]. Yet, IS transpositions can be triggered under severe stress conditions to promote advantageous adaptive mutations, which, in turn, may lead to ecologically significant traits for host fitness. Herein, we reported that IS elements act as intermediaries in the host's fitness trade-off between avoiding genetic predation and acquiring beneficial foreign genes; by disrupting the protective CRISPR-Cas system, ISs may enable the acquisition of MGEs that could significantly contribute to the host's adaptive evolution (Fig. 9).

Our initial screening revealed multiple natural occurrences of various IS transpositions into *cas* genes, potentially disrupting CRISPR-Cas immunity. By employing the highly efficient CRISPR/Cas-mediated IS trapping system, we have successfully characterized a remarkable total of 1457 IS transposition events specifically occurring within

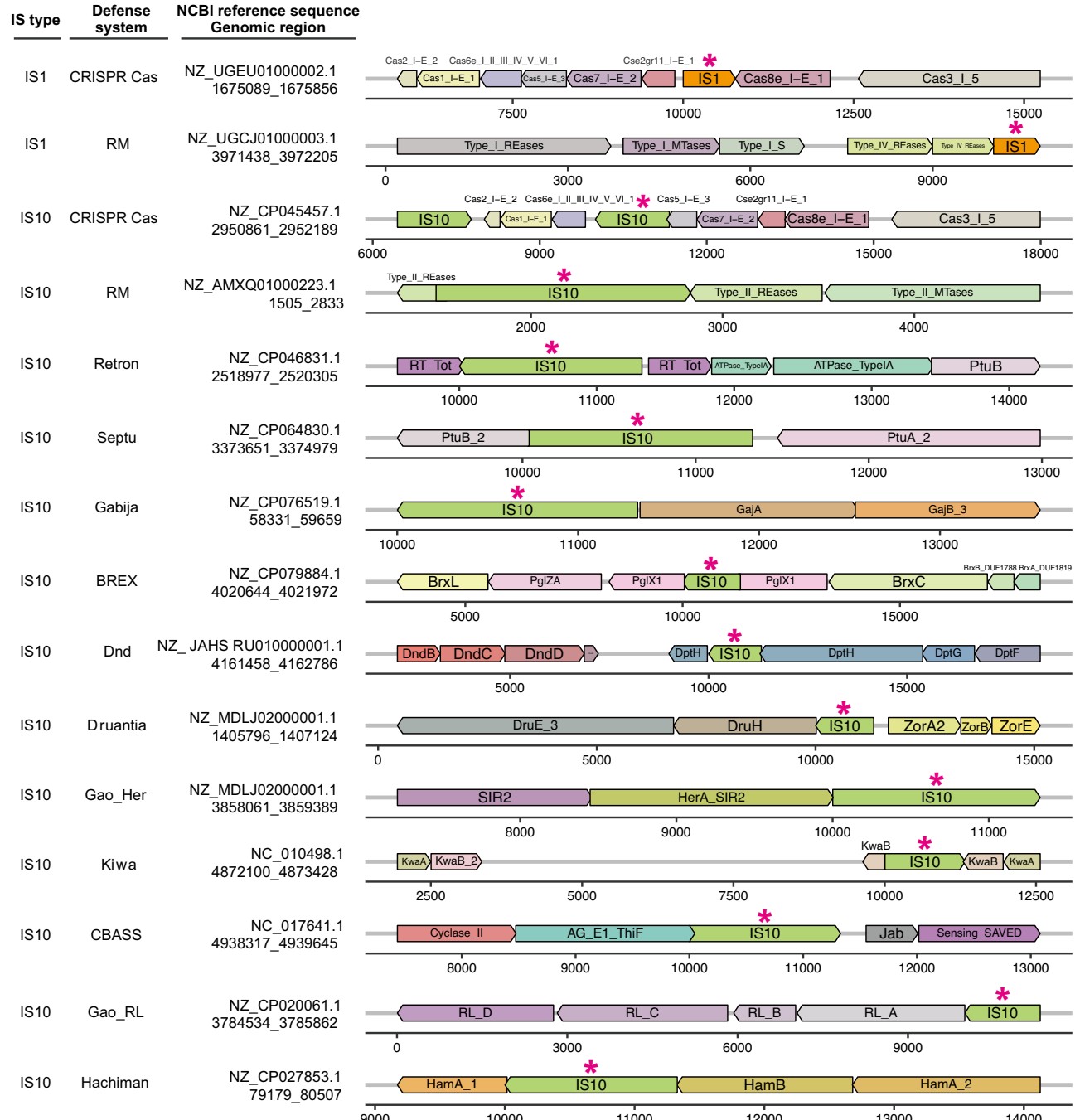

**Fig. 8 | Defense systems compromised by IS1 and IS10 transpositions.** Representative examples of each defense system (listed in second column) are shown. Each row represents a specific transposition event, and the NCBI reference number of each target sequence and coordinates of IS transposition are indicated in the third column. Defense system genes were annotated by DefenseFinder. The orientation of genes and embedded ISs are indicated by colored arrows in either the forward or reverse direction. Horizontal axes indicate the relative position of each gene or IS within the targeted genomic region. For all identified transposition events, see Fig. S22.

various *cas* genes, generating a comprehensive collection of TSD sequences for these ISs. We systematically analyzed the interaction between ISs and CRISPR/Cas systems on a broad scale, uncovering a distinct arsenal for disabling the CRISPR/Cas systems. Currently, only two primary mechanisms that disrupt the proper functioning of CRISPR/Cas systems have been reported. Prophages can disrupt the antiviral functions of CRISPR machinery by integration into the direct repeats of the CRISPR locus, and have evolved anti-CRISPR proteins that can counteract CRISPR/Cas systems (Fig. 9)[45,46]. Compared to these two reported anti-CRISPR mechanisms, ISs, with their pervasive presence and broad target recognition capabilities, emerge as a distinct mechanism for interfering with CRISPR/Cas systems. Remarkably, through iterative mutagenesis of IS target sites in *cas* genes, IS1 and IS10 exhibited greater TSD sequence variability than previously known, and emerged as prominent players in disrupting *cas* genes in *E. coli*, demonstrating their substantial flexibility in target site recognition and greater potential for disrupting defense-associated gene clusters.

Notably, the emergence of IS-mediated transposition events following the introduction of NHEJ machinery indicated that the DNA

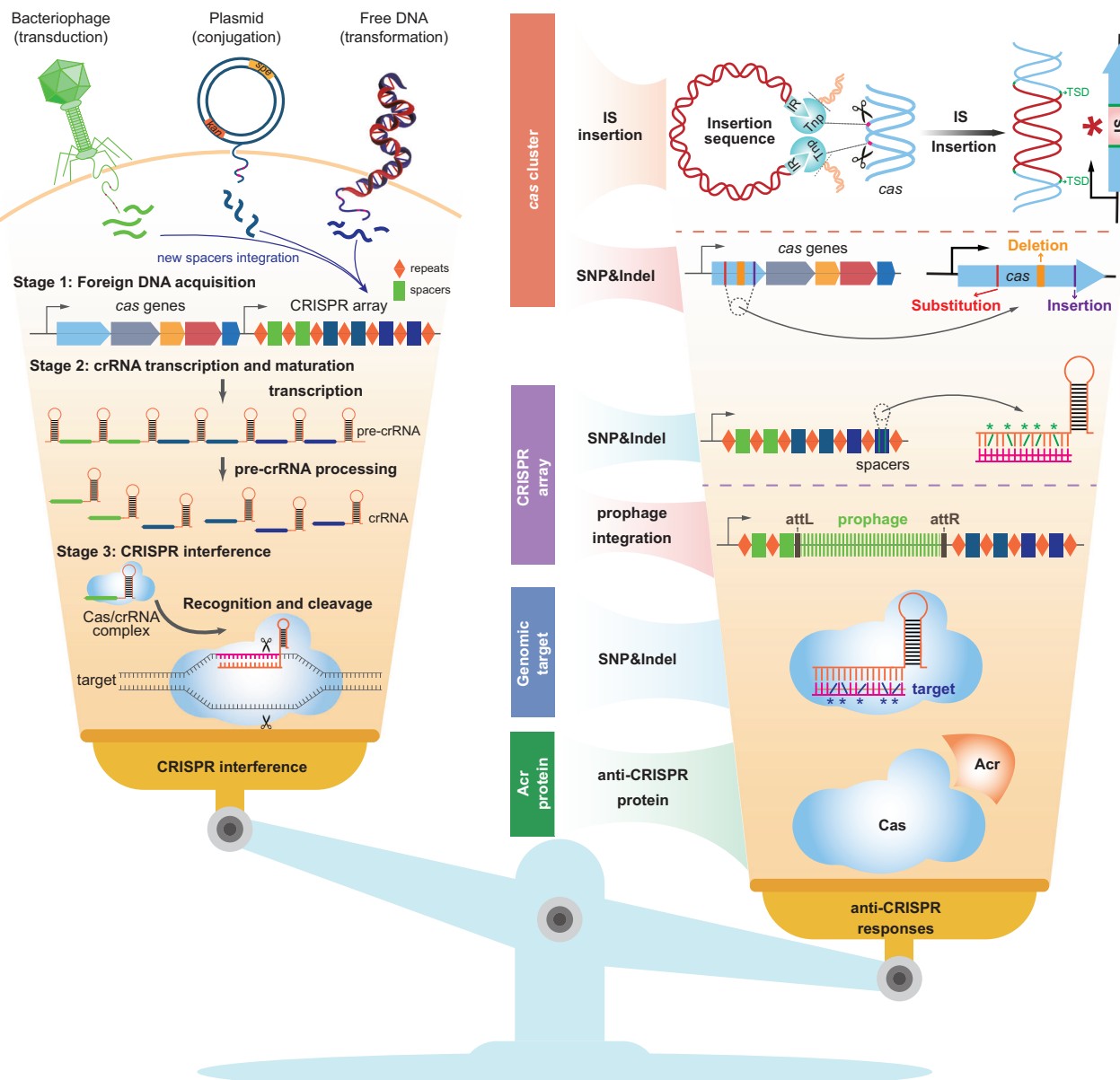

**Fig. 9 | Overview of naturally occurring anti-CRISPR responses.** CRISPR/Cas adaptive immunity consists of three distinct stages: foreign DNA acquisition, crRNA transcription/maturation, and interference (target recognition and cleavage). However, strong selective pressures exerted by CRISPR-Cas has led to the emergence of increasingly powerful anti-CRISPR defenses to combat its cleavage. We discovered that various ISs insert into the *cas* gene, thereby engaging in the host's fitness trade-off (balance) between positive and negative aspects of CRISPR-Cas interference. Also, many bacteriophages have evolved anti-CRISPR (Acr) proteins or site-specific integrases that can disrupt CRISPR-Cas mechanisms, and some single-nucleotide polymorphisms (SNP) and indel mutations in *cas* genes, spacers, or genomic targets can deactivate CRISPR-Cas. Overall, under certain stress conditions, the CRISPR-Cas system may undergo compensatory mutations which can mediate the fitness trade-off between the uptake of beneficial genes and defense against infection.

damage incurred by CRISPR/Cas-mediated cleavage could not be completely repaired by NHEJ alone, underscoring the indispensable role of ISs in mitigating such genotoxicity. In addition, in the CRISPR interference trials, we found escapers with *cas* gene mutations, genomic target alterations, and direct-repeat recombinations in the CRISPR array that are potentially part of the host's intrinsic survival machinery. These multi-level survival mechanisms could sustain the host until the biotic stress, such as the DSB-induced genotoxicity in our study, is eliminated.

The pervasiveness of IS elements in diverse organisms suggests that their transposition abilities are indispensable for the adaptation and biological diversification of the hosts. Notably, we found that IS elements can potentially breach a variety of host defense systems,

making ISs key players in the ongoing evolutionary competition between genetic integrity and acquisition of potentially favorable MGEs.

Moreover, our findings have also uncovered the inherent risk posed by chromosome-encoded ISs, as they are capable of impairing the proper functioning of CRISPR/Cas systems, resulting in numerous escapers during genome editing. This work underscores the need for careful considerations regarding host selection to minimize the impact of ISs-mediated escape events, thereby ensuring the desired genome-editing efficiency. In the case of *E. coli*, a judicious choice would be the IS-free MDS42 strain[47] for gene editing, thereby eliminating any gene editing escape events arising from ISs transpositions into *cas* genes and ensuring successful editing outcomes. The discovery of such anti-

CRISPR mechanism highlights the importance of considering not just the off-target effects but also the potential disruptive effects of host-encoded ISs on CRISPR/Cas functionality during gene editing, thereby optimizing gene editing efficiency and accelerating the further applications of CRISPR/Cas systems.

Together, our findings have broad implications for understanding how IS transposition drives the host to neutralize various immunization barriers. Such "ingenious" MGEs, represented especially by IS1 and IS10, display notable flexibility through their substantial compatibility with varied TSD sequences, potentially endowing their hosts with survival advantages. Further exploration of the evolutionary contribution of ISs will help not only in understanding of ISs for driving environmental adaption of prokaryotes, but in technological applications, including strain development and genome editing.

## Methods

### Primers, plasmids, bacterial strains, and growth conditions

Primers, plasmids, and strains used in this study are listed in Tables S1–S3, respectively.

*Escherichia coli* strains MG1655, DH10B, MDS42, and their derivates were cultured in LB medium at 37 °C, with shaking at 220 rpm. All the recombinants containing the temperature-sensitive pSC101-derived plasmid were cultured at 30 °C. When necessary, appropriate antibiotics were added at the following concentrations: tetracycline, 5 μg/mL; spectinomycin, 50 μg/mL; chloramphenicol, 25 μg/mL; kanamycin, 50 μg/mL; apramycin, 50 μg/mL; and ampicillin, 100 μg/mL. In addition, 0.5 mM isopropyl β−ᴅ-1-thiogalactopyranoside (IPTG) or 10 mM ʟ-arabinose was added into the broth to appropriately induce the promoter pTrc or pBAD initiation, respectively.

### Construction of *recA*-deficient strains

The *recA*-deficient knock-out mutants (*E. coli* MG1655-ΔrecA and MDS42-ΔrecA) harboring the Flp recombination target (FRT) scar were generated through the FLP/FRT system as previously described[48].

### Assembly cloning of CRISPR-Cas locus from *Erwinia amylovora* 7-3 into *E. coli* DH10B

Based on the CRISPRCasFinder[49], we determined that *E. amylovora* 7-3 possessed a canonical type I-E CRISPR-Cas system composed of two CRISPR arrays (CRISPR 1 and CRISPR 2) and a *cas* gene cluster. Notably, endogenous Cas7 expression was potentially disrupted by IS10 transposition. To reconstruct the functional type I-E CRISPR-Cas system of *E. amylovora* 7-3, we removed the IS10 element and one of the resulting 9 bp TSD sequences from the CRISPR-Cas locus. Additionally, we filtered out most spacers that could not target the plasmid p7-3 from two CRISPR arrays to achieve a more accurate assessment of CRISPR interference against p7-3. Our artificial CRISPR-Cas locus (EraCas) spanned ~11 kb and consisted of eight *cas* genes, two shortened CRISPR arrays, in addition to the potential leader sequences. The constructs were generated as indicated below.

The entire EraCas region was further divided into five segments and then synthesized de novo by the Beijing Qingke Biotechnology Co., Ltd. and separately cloned into plasmid pClone007 to yield plasmids pClone007-EraCas01, pClone007-EraCas02, pClone007-EraCas03, pClone007-EraCas04 and pClone007-EraCas05. The three partially overlapping PCR fragments EraCas02, EraCas03 and EraCas04 were amplified from the corresponding synthetic templates with primer pairs EraCas-locus234-2-F/R, EraCas-locus234-3-F/R and EraCas-locus234-4-F/R, respectively, and then assembled by overlap PCR with primer pair EraCas-locus234-2-F/EraCas-locus234-4-R. The resulting amplicon was directly ligated into the *Eco*RV site of pBlue-Script II SK + , generating plasmid pEraCas234. Afterwards, the assembled fragment EraCas234 was double digested with *Bgl*II/*Hin*dIII and subsequently ligated into plasmid pClone007-EraCas01, which was similarly digested, yielding plasmid pEraCas1234. The final

fragment, EraCas05, obtained from pClone007-EraCas05, was digested with *Hin*dIII/*Xho*I and ligated into the *Hin*dIII/*Xho*I cloning site of pEraCas1234 to generate pEraCas, which included an ~11 kb engineered type I-E CRISPR-Cas locus of *E. amylovora* 7-3.

Additionally, we engineered an IS10-aborted CRISPR-Cas gene cluster (EraCas-IS10), designed from the *E. amylovora* 7-3 genomic sequence, as the negative control. The IS10 element and one of the resulting TSD sequences (TGCGCACCA) were separated into two segments and individually amplified from *E. coli* DH10B with primer pairs T-Apr-IS10-Era-1-F/R and T-Apr-IS10-Era-3-F/R, and the apramycin selective marker was amplified from plasmid pSET152 using primer pairs T-Apr-IS10-Era2-F/R. The resulting three partially overlapping fragments were assembled by overlap PCR with primer pair T-Apr-IS10-Era-1-F/T-Apr-IS10-Era-3-R and subsequently recombined with plasmid pEraCas by λ-red-mediated recombination, yielding plasmid pEraCas-IS10-Apr. The apramycin resistance cassette was removed from pEraCas-IS10-Apr through *Pag*I digestion and self-ligation, yielding the desired control plasmid pEraCas-IS10.

The engineered protospacers, designed from plasmid p7-3, were synthesized de novo by the Beijing Qingke Biotechnology Co., Ltd. and cloned into a p15A-derived plasmid to produce p15A-Eraprotos. The spectinomycin resistance cassette, obtained from SK(+)-spe (previously constructed), was digested with *Kpn*I/*Not*I and then cloned into similarly digested p15A-Eraprotos to generate p15A-Eraprotos-spe. All protospacers were deleted from p15A-Eraprotos-spe by *Mau*BI digestion and self-ligation, yielding the negative control plasmid p15A-spe, which was not susceptible to the type I-E CRISPR-Cas system of *E. amylovora* 7-3.

The *araC* gene and pBAD promoter were amplified from plasmid pCas (Addgene plasmid # 62225), which was kindly provided by Prof. Sheng Yang from the Institute of Plant Physiology and Ecology (Shanghai, China), using primer pair T-Apr-pBAD-2-F/R, and the apramycin selective marker was amplified from plasmid pSET152 using primer pair T-Apr-pBAD-1-F/R. The resulting two partially overlapping fragments were assembled by overlap PCR with primer pair T-Apr-pBAD-1-F/T-Apr-pBAD-2-R and subsequently recombined with plasmid pEraCas by λ-red-mediated recombination, generating plasmid pEraCas-pBAD, which harbored the ʟ-arabinose-inducible pBAD promoter to drive expression of the *cas* operon.

All protospacers designed from p7-3, together with the chloramphenicol resistance cassette, were amplified from p15A-Eraprotos-spe using primer pair T-EraP-pro-Ichl-F/R and inserted into the DH10B chromosome by λ-red-mediated recombination, yielding strain SYHY03, which could be efficiently targeted by pEraCas-pBAD in the presence of ʟ-arabinose.

### Analysis of growth curves

Individual transformants were cultivated overnight at 37 °C in LB medium with appropriate antibiotics, followed by 1:100 dilutions and inoculation into a Bioscreen 100-well honeycomb plate containing 400 μL LB broth with different antibiotic concentrations. The OD600 was monitored every 20 min for 25 h at 37 °C with continuous shaking in a Bioscreen C apparatus (Bioscreen C Labsystems).

### Analysis of plasmid stability

All of the candidate strains harboring the evaluated plasmids were grown overnight at 37 °C in LB broth with appropriate antibiotics (ampicillin, spectinomycin, and chloramphenicol), followed by 1:100 dilutions and inoculation into LB medium containing only ampicillin, which was used for the maintenance of plasmid pEraCas. Cells were routinely passaged every 12 h in the absence of spectinomycin and chloramphenicol for up to 3 passages. Following serial dilutions, each culture solution was plated on ampicillin (A) LB agar plates, with and without spectinomycin-chloramphenicol (SC), for the determination of the colony-forming units (CFUs). Plasmid stability was calculated by

dividing the number of CFU on the ASC plate by the number of CFU on the A plate.

### Reverse transcription quantitative PCR analysis

An overnight culture of DH10B containing pEraCas or pEraCas-pBAD was inoculated into LB medium containing appropriate antibiotics and grown at 37 °C. After the OD600 reached 0.25–0.30, 10 mM L-arabinose was added to induce the expression of the *cas* operon within pEraCas-pBAD for 2.5 h. The cells were then centrifuged and collected, and total RNA was extracted with the RNAprep Pure Cell/Bacteria Kit (Tiangen Biotech Beijing, China), followed by digestion of genomic DNA with DNaseI. In all, 500 ng total RNA was subsequently reverse transcribed into cDNA using the Hifair® III 1st Strand cDNA Synthesis Kit (gDNA digester plus) (YeaSen Biotech, Shanghai, China). The qPCR was performed on the qTOWER3G system (Analytik Jena, Germany) with the Hieff™ qPCR SYBR® Green Master Mix (Low Rox Plus) (YeaSen Biotech, Shanghai, China), and the relative expression levels of *cas* genes were calculated using the formula $2^{-\Delta\Delta Ct}$. A list of gene-specific primers for RT-qPCR can be found in Table S1.

### Construction of CRISPR interference system-associated plasmids

The plasmid pCas was used as the original vector backbone for further construction. The tetracycline-resistance cassette, flanked by short homology arms, was amplified from plasmid pSC101 with primers Tar-tet-F/R and inserted into pCas to obtain the plasmid pCas-tet using the λ-red-mediated homologous recombination. Next, the spectinomycin-resistance cassette and pTrc promoter were individually amplified using primer pairs Tar-SmR-pTrc-1-F/R and Tar-SmR-pTrc-2-F/R, respectively, and then the amplicons were ligated together by overlap PCR with primers Tar-SmR-pTrc-1-F and Tar-SmR-pTrc-2-R. The resulting PCR product was recombined with pCas-tet to generate plasmid pCasY-1 by replacing the kanamycin-resistance cassette with the amplified cassette, thus driving the expression of SpCas9 from the pTrc promoter.

To reduce the off-target effects of the CRISPR-Cas system, the N497A, R661A, Q695A, and Q926A multiple-site mutations were simultaneously introduced in the coding region of SpCas9 within pCasY-1 by site-directed mutagenesis with five overlapping primer pairs (SpCas9-HF1-1-F/R, SpCas9-HF1-2-F/R, SpCas9-HF1-3-F/R, SpCas9-HF1-4-F/R, and SpCas9-HF1-5-F/R); the resulting plasmid pCasY-2 contained the high-fidelity SpCas9-HF1 nuclease with the ssrA degradation tag fused at its C-terminus.

Furthermore, to eliminate interference from the λ-red recombination system of pCasY-2, plasmids pCasY-3-ΔSIM-kan (without the ssrA degradation tag at the C-terminus of SpCas9-HF1) and pCasY-3-ΔSIM-kan-ssrA (with the ssrA degradation tag fused to the C-terminus of SpCas9-HF1) were separately constructed by substituting the λ-red expression cassette with the kanamycin-resistance cassette amplified from the plasmid SuperCos 1 using primers Tar-Δλ-red-kan-F/R and Tar-Δλ-red-kan-ssrA-F/R, respectively. One of the resulting plasmids, pCasY-3-ΔSIM-kan-ssrA, was then digested by the restriction endonuclease *Pvu*I to specifically delete both the C-terminus of the spectinomycin-resistance protein coding region and the complete expression cassette of SpCas9-HF1-ssrA, followed by in vitro self-ligation and transformation to generate the recirculated vector backbone pCasY-3-ΔSIM-kan-ssrA-PvuI for subsequent cloning steps.

For the construction of plasmid pCpf1Y-3-ΔSIM-kan harboring the CRISPR-Cpf1 system, the fragment containing both the complementary C-terminus of the spectinomycin-resistance protein coding region and the pTrc promoter was amplified from pCasY-2 with primers Tar-spe-FnCpf1-1-F/R, and the FnCpf1 coding region from plasmid pY001 (Addgene plasmid #69973) was amplified with primers Tar-spe-FnCpf1-2-F/R. Using primers Tar-spe-FnCpf1-1-F and Tar-spe-FnCpf1-2-R, the two amplicons were ligated together through overlap

PCR to further facilitate homologous recombination with the pCasY-3-ΔSIM-kan-ssrA-PvuI vector backbone.

### Development of a dual CRISPR nickase strategy

The mutations D10A and H840A were individually introduced in the coding region of SpCas9-HF1 within the plasmid pCasY-3-ΔSIM-kan by site-directed mutagenesis with overlapping primers Tar-spe-SpCas9-HF1(D10A)-1-F/R and Tar-spe-SpCas9-HF1(D10A)-2-F/R (for mutation D10A), and Tar-spe-SpCas9-HF1(H840A)-1-F/R and Tar-spe-SpCas9-HF1(H840A)-2-F/R (for mutation H840A), resulting in the nickase plasmids pCasY-3-ΔSIM-kan (D10A) and pCasY-3-ΔSIM-kan (H840A), respectively. Additionally, the negative control plasmid pCasY-3-ΔSIM-kan (D10A&H840A) encoding the catalytically inactive SpCas9-HF1 was constructed by incorporating the double mutations D10A and H840A into the coding region of SpCas9-HF1 within pCasY-3-ΔSIM-kan using overlap-extension PCR with primers Tar-spe-SpCas9-HF1(D10A)-1-F/R, Tar-spe-SpCas9-HF1(D10A)-2-F/Tar-spe-SpCas9-HF1(H840A)-1-R, and Tar-spe-SpCas9-HF1(H840A)-2-F/R. The subsequent combination of the mutant nickases with a single or a pair of sgRNAs would result in single-strand breaks (SSBs) or double-strand breaks (DSBs), respectively.

### Design and cloning of the appropriate sgRNAs and crRNAs for CRISPR interference

Both sgRNAs (for SpCas9-HF1) and crRNAs (for FnCpf1) were designed to target the four non-essential genomic genes *pyrF*, *lpp*, *ompF*, and *lacZ*, using the CRISPOR algorithm (http://crispor.tefor.net)[50]. Additionally, to create specific DSBs at the *ompF* coding region for the mutant nickase versions of SpCas9-HF1 (D10A) and SpCas9-HF1 (H840A), two pairs of sgRNAs with different offsets were designed by the CHOPCHOP program (http://chopchop.cbu.uib.no)[51]. These targeting cassettes, together with two non-targeting sequences (negative controls), were synthesized de novo by Beijing Qingke Biotechnology Co., Ltd., and then cloned into the vector backbone p15A-cm to obtain all the targeting plasmids.

### Development of the CRISPR interference system

All pre-constructed CRISPR-Cas/Cpf1 expression plasmids were separately electroporated (at 25 μF, 200 Ω, and 1.8 kV) into the appropriate hosts, then immediately resuspended in fresh LB medium and allowed to recover at 30 °C for 1 h, followed by plating on LB agar plates supplemented with 5 μg/mL tetracycline, 50 μg/mL spectinomycin, and 50 μg/mL kanamycin, and then incubated at 30 °C for 24 h. Individual colonies were selected and cultured with appropriate antibiotics for the subsequent CRISPR interference experiments. Afterwards, the electrocompetent cells harboring the CRISPR-Cas/Cpf1 expression plasmids were prepared, and then transformed with 100 ng sgRNA or crRNA expression plasmids without any DNA donor template. The recovered cells were subsequently grown at 30 °C for 36 h on LB agar plates with 5 μg/mL tetracycline, 50 μg/mL spectinomycin, 50 μg/mL kanamycin, 25 μg/ mL chloramphenicol, and 0.5 mM IPTG. The single colonies from each plate were selected randomly for colony PCR with the primer pair S-univer-c&9-F/R to amplify the complete coding regions of SpCas9-HF1, FnCpf1, and their derivatives. PCR products were detected by 0.6% agarose gel electrophoresis, followed with gel purification and Sanger sequencing.

### Removal of potential IS target sites in the coding sequences of SpCas9-HF1 and FnCpf1 through iterative mutation strategy

In brief, based on codon degeneracy, global reconstruction of nucleotide sequences was firstly performed on the coding regions of SpCas9-HF1 and FnCpf1, followed by two sequential rounds of site-directed mutagenesis, to exclude the potential target sites of IS1, IS5, and IS10 as much as possible. In the first reconstruction round, each of the full-length coding sequences of SpCas9-HF1-ssrA and FnCpf1 was

globally shuffled to eliminate the TSD motifs of IS1 (IS1-8: NRAWWWWN), IS5 (IS5: YTAR), and IS10 (IS10: NRCWNWRYN), together with all repeats greater than or equal to 10 nucleotides using the software "Gene Designer" (https://www.atum.bio/resources/tools/gene-designer)[52]. Each modified sequence was further divided into three segments and then synthesized de novo by General Biological System (Anhui) Co., Ltd. (Anhui, China).

The three partially overlapping PCR fragments SpCas9-HF1-OP01-01, SpCas9-HF1-OP01-02, and SpCas9-HF1-OP01-03 were amplified from the corresponding synthetic templates with primer pairs Tar-spe-CasOP01-1-F/R, Tar-spe-CasOP01-2-F/R, and Tar-spe-CasOP01-3-F/R, respectively. In the same way, fragment SpCas9-HF1-OP01-ssrA-03 (with the ssrA degradation tag fused at the C-terminus of SpCas9-HF1-OP01) and the overlapping PCR products FnCpf1-OP01-01, FnCpf1-OP01-02, and FnCpf1-OP01-03 were amplified by PCR with primer pairs Tar-spe-CasOP01-3-F/Tar-spe-CasOP01-ssrA-3-R, Tar-spe-FnCpf1OP01-1-F/R, Tar-spe-FnCpf1OP01-2-F/R, and Tar-spe-FnCpf1OP01-3-F/R, respectively. In addition, the DNA segment spe-pTrc-HRCasOP01, which contains the partial C-terminus of the spectinomycin-resistance protein coding region, the promoter pTrc, and the homologous arm HRCasOP01, was amplified from plasmid pCasY-2 by PCR with primers Tar-spe-F/Tar-spe-CasOP01-R. The DNA segment spe-pTrc-HRFnCpf1OP01 was amplified by PCR with primers Tar-spe-F/Tar-spe-Cpf1OP01-R in the same way. Subsequently, the amplified, partially overlapping PCR products spe-pTrc-HRCasOP01, SpCas9-HF1-OP01-01, SpCas9-HF1-OP01-02, and SpCas9-HF1-OP01-03 were assembled via overlap PCR with primers Tar-spe-F/Tar-spe-CasOP01-3-R to generate the PCR product Tar-spe-CasOP01. Similarity, the PCR products spe-pTrc-HRCasOP01, SpCas9-HF1-OP01-01, SpCas9-HF1-OP01-02, and SpCas9-HF1-OP01-ssrA-03 were joined together using overlap PCR with primers Tar-spe-F/Tar-spe-CasOP01-ssrA-3-R to obtain the PCR product Tar-spe-CasOP01-ssrA. Additionally, the PCR products spe-pTrc-HRFnCpf1OP01, FnCpf1-OP01-01, FnCpf1-OP01-02, and FnCpf1-OP01-03 were joined by overlap PCR with primers Tar-spe-F/Tar-spe-FnCpf1OP01-3-R to yield the PCR product Tar-spe-FnCpf1OP01.

The resulting recombinant PCR products Tar-spe-CasOP01, Tar-spe-CasOP01-ssrA, and Tar-spe-FnCpf1OP01 were separately cloned into the vector backbone pCasY-3-ΔSIM-kan-ssrA-PvuI using the λ-red homologous recombination system to obtain plasmids pCasY-3-ΔSIM-kan-OP01, pCasY-3-ΔSIM-kan-OP01-ssrA (with the ssrA degradation tag fused at the C-terminus of SpCas9-HF1-OP01), and pCpf1Y-3-ΔSIM-kan-OP01, respectively.

Finally, the last two sequential rounds of site-directed mutagenesis were achieved by successive rounds of overlap PCR with primers listed in Table S1 and standard λ-red recombinant techniques. The resulting plasmids were designated pCasY-3-ΔSIM-kan-OP02, pCasY-3-ΔSIM-kan-OP02-ssrA, pCasY-3-ΔSIM-kan-OP03, pCasY-3-ΔSIM-kan-OP03-ssrA, pCpf1Y-3-ΔSIM-kan-OP02, and pCpf1Y-3-ΔSIM-kan-OP03.

### Cloning and assembly of NHEJ expression cassettes
To introduce the non-homologous end-joining (NHEJ) repair pathways from three different strains, including *Mycobacterium smegmatis* mc²155, *Mycobacterium marinum* M, and *Bacillus subtilis* 168, into *E.coli* MG1655-ΔrecA, the DNA ligase D gene (*ligD*) from each strain, an additional promoter J23119 and the gene *ku* were synthesized de novo by the Beijing Qingke Biotechnology Co., Ltd. and the General Biological System (Anhui) Co., Ltd. (Anhui, China), and then separately inserted into the cloning vector pUC57-1.8 K to obtain plasmids pUC57-1.8K-Ms-ligD, pUC57-1.8K-Ms-ku, pUC57-1.8K-Mm-ligD, pUC57-1.8K-Mm-ku, pUC57-1.8K-Bs-ligD, and pUC57-1.8K-Bs-ku. Subsequently, the complete coding regions of Mm-ku, Ms-ku and Bs-ku were amplified by PCR from plasmids pUC57-1.8K-Mm-ku, pUC57-1.8K-Ms-ku, and pUC57-1.8K-Bs-ku with primers pBAD-Mm-ku-2-F/R, pBAD-Ms-ku-2-F/R, and pBAD-Bs-ku-2-F/R, respectively, and individually placed

downstream of the pBAD promoter by overlap PCR with primers pBAD-Mm-ku-1-F/-2-R, pBAD-Ms-ku-1-F/-2-R, and pBAD-Bs-ku-1-F/-2-R, respectively. Each assembled PCR product was subsequently ligated into the *Eco*RV site of the cloning plasmid pBluescript SK(+) to generate plasmids SK(+)-pBAD-Mm-ku, SK(+)-pBAD-Ms-ku, and SK(+)-pBAD-Bs-ku, respectively. In addition, each pBAD-ku expression cassette from SK(+)-pBAD-Mm-ku, SK(+)-pBAD-Ms-ku, and SK(+)-pBAD-Bs-ku was doubly digested with restriction enzymes *Not*I/*Bcu*I, *Bcu*I/*Psc*I, and *Not*I/*Bcu*I, respectively, and then separately cloned into plasmids pUC57-1.8K-Mm-ligD, pUC57-1.8K-Ms-ligD, and pUC57-1.8K-Bs-ligD to generate the final NHEJ expression plasmids pMm-NHEJ, pMs-NHEJ, and pBs-NHEJ, respectively.

### Establishment of the NHEJ repair system assay
To assess NHEJ repair of CRISPR/Cas-mediated DSBs, competent *E. coli* MG1655-ΔrecA cells harboring the SpCas9-HF1 nuclease expression plasmid (pCasY-3-ΔSIM-kan), along with each type of NHEJ expression plasmid (pMm-NHEJ, pMs-NHEJ, or pBs-NHEJ), were separately prepared with or without the 10 mM L-arabinose induction, and subsequently electroporated with 100 ng sgRNA expression plasmid (p15A-cm-sgRNA::ompF). The recovered cells were then plated and incubated at 30 °C for 36 h on LB agar plates containing 5 μg/mL tetracycline, 50 μg/mL spectinomycin, 50 μg/mL kanamycin, 25 μg/mL chloramphenicol, and 0.5 mM IPTG. Indel mutations within the target site (*ompF* gene) of survival cells (escapers) were detected as described below.

### Determination and visualization of indel mutations generated by NHEJ repair
The *ompF* genes of the evaluated strains were amplified by PCR with primers V-ompF-F/R, and the gel-purified PCR products were then subjected to Sanger sequencing to validate the existence of mutation sites which were generated by NHEJ repair. Alignments of the sequencing results were performed using the algorithm MUSCLE (available at https://www.drive5.com/muscle/)[53] with the default settings, and subsequently displayed with the R-package "ggmsa" (available at https://anaconda.org/bioconda/r-ggmsa)[54].

### Analysis and identification of naturally occurring transpositions of ISs into CRISPR-Cas gene clusters
The FASTA sequences of IS elements can be found in the ISfinder database (https://www-is.biotoul.fr). The CRISPRCasdb database containing the identified CRISPR-Cas systems from complete genome sequences was downloaded and further parsed on the Linux operating system. Based on the GenBank assembly accession numbers extracted from the CRISPRCasdb database, the genomic sequences harboring CRISPR-Cas gene clusters were downloaded from NCBI. According to the genomic coordinate data, the CRISPR-Cas gene cluster sequences were subsequently extracted from the downloaded genomic sequences, and merged into a single database for analysis. A local BLASTn search of ISs against CRISPR-Cas gene clusters was performed with a cut-off of 100% sequence identity and 100% coverage. The graphical representations of candidate CRISPR-Cas gene clusters were visualized using the R package gggenes (available at https://CRAN.R-project.org/package=gggenes).

### Identification of putative spacer-protospacer matches
A collection of the plasmid sequences was retrieved from the NCBI Nucleotide database (accessed March 2022) by filtering the genetic compartments using the key word "Plasmid". Spacers of the CRISPR loci containing IS insertions into *cas* genes were predicted by CRISPRCasFinder, and subsequently subjected to BLASTn against the plasmid database with a cut-off of 100% nucleotide identity and 100% coverage. To eliminate the false-positive prediction of spacers, the matches were then manually corrected.

### Detection and initial annotation of open reading frames

Identification of open reading frames and initial annotations were rapidly conducted with the "Prokka" annotation pipeline (available at https://github.com/tseemann/prokka)[55] using default settings, and the resulting *.gff and *.faa format files containing all putative genes were used for further analysis.

### Identification of antimicrobial resistance genes

All nucleotide sequences of the candidate plasmids were firstly subjected to Prokka annotation to obtain the coding sequences. Afterwards, using the resulting *.faa format file as input, antimicrobial resistance genes were detected by the online ResFinder server[56].

### Whole-genome sequencing analysis of CRISPR-tolerant mutants

All test strains were cultured in 5 mL LB medium containing appropriate antibiotics at 30 °C, with shaking at 220 rpm overnight. Following three washes in ice-cold phosphate-buffered saline, the cells were collected by centrifugation and then fast-frozen in liquid nitrogen. Subsequent genomic DNA extraction, library construction and whole-genome sequencing were conducted by Sinotech Genomics Co. Ltd. (Shanghai, China). All genomes were sequenced by a combination of Illumina NovaSeq6000 with Pacific Biosciences Sequel II or Oxford Nanopore PromethION platforms. A detailed description of these samples was available in Table S6. Through the NUCmer algorithm from MUMmer suite[57], each complete chromosome sequence was then individually aligned to the control sequence to identify genetic changes.

### Statistical analysis of IS element diversity and relative abundance

Unique genomic sequences (60,616 archaea; 19,589,090 bacteria; and 14,912 viruses) were programmatically downloaded from the NCBI RefSeq database (accessed February 2022) through the executable software Aspera (https://developer.asperasoft.com/desktop-advance/command-line-client) and ncbi-genome-download scripts (https://github.com/kblin/ncbi-genome-download/). Alignments of ISs and genomic sequences were performed using BLASTn, and matches with 100% nucleotide identity and 100% coverage were retained. Taxonomy lineage for each identified genomic sequence was retrieved from the NCBI taxonomy database using TaxonKit tool (available at https://github.com/shenwei356/taxonkit)[58] and then further classified into 6 levels, including kingdom, phylum, class, order, family, and genus. After taxonomic assignments, the fraction of genomes harboring each IS family in each taxonomic category was calculated and visualized by R package "pheatmap" (available at https://CRAN.R-project.org/package=pheatmap). Additionally, the total number of IS families per genome was computed for each taxonomic category and then plotted in boxplot format by the "ggplot2" package in RStudio (available at https://ggplot2.tidyverse.org).

### Determination of the target site duplications

The S-univer-c&9-F/R primer pair was designed to amplify the longest PCR products spanning the complete coding regions of SpCas9-HF1, FnCpf1, and their derivatives. All PCR products were subsequently purified and sequenced by the Sanger method with two IS-specific reverse primers. On the basis of the sequencing results, we manually determined the target site duplications (TSDs) for each IS element.

Additionally, the potential TSDs generated by IS1 and IS10 within all naturally occurring transposition events were analyzed utilizing an array of bioinformatic tools. A single FASTA file containing the nucleotide sequences of IS1 and IS10 from *E. coli* MG1655 and *E. coli* DH10B was created for downstream analysis. A local BLASTn search with a 100% sequence identity and 100% coverage threshold was performed to match both IS1 and IS10 with the unique genomic sequences previously downloaded from the NCBI RefSeq database. Based on the

output coordinates of IS1 and IS10, the flanking regions spanning 8 bp and 9 bp for IS1 and 9 bp for IS10, upstream and downstream of each specific genome sequence, were extracted by Seqkit tool with the command "seqkit-subseq" (available at http://bioinf.shenwei.me/seqkit)[59]. To further investigate the possible presence of TSDs resulting from IS1 and IS10 transpositions, the sequences immediately adjacent to the element boundaries were detected by the pattern search program PatScan (available at https://patscan.secondarymetabolites.org)[60] with the following specific computed parameters: "p1=NNNNNNNN p2 = 768...768 p3 = p1" for IS1-8 (the length of TSD was 8 bp), "p1=NNNNNNNNN p2 = 768...768 p3 = p1" for IS1-9 (the length of TSD was 9 bp) and "p1=NNNNNNNNN p2 = 1329...1329 p3 = p1" for IS10, respectively.

### Generation and visualization of TSD motifs

All identified TSDs were pre-clustered and screened to generate the single multi-sequence FASTA files according to the IS type and the TSD length. The position frequency matrix (PFM) was computed in each FASTA file in RStudio by determining the frequency of each nucleotide at each position within the TSD and then used as input into the Bioconductor R package "motifStack" (available at https://bioconductor.org/packages/motifStack/)[61] to generate the motif logo.

### Comparative analysis of IS transposition spectrum diversity

The obtained TSDs were firstly mapped to the target genes using the Seqkit algorithm, and sequences with exact matches were then carefully examined to eliminate any false positive loci resulting from the off-target sequences with multiple matches with TSDs. A matrix was created, with rows corresponding to genes, columns corresponding to a collection of IS insertion sites, and entries corresponding to whether transposition occurred at this locus for each IS element. Finally, a comparative analysis of IS transposition spectrum diversity across various genes was conducted and plotted using OncoPrint within the Bioconductor R package "ComplexHeatmap" (available at https://bioconductor.org/packages/ComplexHeatmap/)[62].

### Exploration of the defense systems disrupted by IS1 and IS10 transpositions

The genomic coordinates of IS1 and IS10 transpositions were obtained as described above, and then used in combination with the Seqkit tool using the command (seqkit subseq --bed *. file -u 10, 000 -d 10, 000) to extract target regions extending 10 kbp upstream and downstream from each IS element. Afterwards, the protein-coding regions of the resulting nucleotide sequences were predicted and annotated by Prokka, followed by searching against the 60 families of antiviral systems in DefenseFinder with the default settings. Finally, the acquired relative coordinates of both ISs and defense genes were imported into the R package "gggenes" for further analysis and visualization.

### Statistical information

All data analysis and statistic were done in R 4.2.2 using RStudio. Statistical details of experiments, including the number of replicates, statistical significance, and statistical test are indicated in the relevant figure legends. Both the GraphPad Prism (version 9.5.1) and RStudio were used to plot figures. The graphs were then modified in Adobe Illustrator (version 26.1).

### Reporting summary

Further information on research design is available in the Nature Portfolio Reporting Summary linked to this article.

## Data availability

All relevant data are included in the paper and/or its supplementary information files. Source data are provided as a Source Data file. All whole-genome sequencing data of CRISPR-tolerant mutants were

deposited in the National Center for Biotechnology Information (NCBI) Sequence Read Archive (SRA) repository under project number PRJNA884016, and the assembled chromosome sequences were also submitted to the GeneBank with accession numbers given in Table S6. Source data are provided with this paper.

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

## Acknowledgements

This work was supported by the National Key R&D Program of China [2018YFA0901900 (Q.K. and Y.O) and 2021YFC2100600 (Y.O and L.B)], the "Major Project" of Haihe Laboratory of Synthetic Biology [22HHSWSS00001 (Q.K., M.T., and S.L.)] and Shanghai Jiao Tong University Medical-Engineering Cross Fund [YG2019QNA53 (Q.K.) and YG2022QN071 (Q.K.)]. In particular, the authors would like to thank Dr. Susan T. Howard for valuable suggestions and discussions. The authors would like to thank for data collection in the Core Facilitiy and Service Center (CFSC) for School of Life Sciences and Biotechnology, SJTU. The computations in this paper were run on the π 2.0 cluster supported by the Center for High Performance Computing at Shanghai Jiao Tong University.

## Author contributions

Conceptualization by Y.S., H.W., and Q.K.; Methodology by Y.S., H.W., and Q.K.; Software by Y.S.; Validation by Y.S. and H.W.; Formal Analysis by Y.S., H.W., and Q.K.; Investigation by Y.S., and H.W.; Data curation by Y.S., H.W., and Q.K.; Writing-original draft by Y.S., H.W., Y.O., Z.D., L.B., and Q.K.; Writing-review and editing by Y.S., H.W., Y.O., Y.W., W.D., Z.D., L.B., and Q.K.; Visualization by Y.S.; Supervision by Y.W., W.D., S.L., M.T., Z.D., L.B., and Q.K.; Funding acquisition by Y.O., S.L., M.T., Z.D., L.B., and Q.K.

## Competing interests

The authors declare no competing interests.
