## [Peer Review File · Nature Communications]

Insertion sequence transposition inactivates CRISPR-Cas immunityReviewer #1 (Remarks to the Author):

A fascinating and rapidly advancing area of study in bacteriology is focused on unraveling how various mobile genetic elements collaborate to drive bacterial evolution. In this manuscript, the authors delve into the intricate interplay between insertion sequences (ISs) and the CRISPR-Cas system. Their proposed hypothesis suggests that ISs can potentially disrupt the activity of the CRISPR-Cas system, thereby increasing the ability of recipient cells to acquire foreign DNA.

While this idea is interesting, this reviewer feels that the manuscript is in its early stages, and additional experiments are required to support this hypothesis. Specifically, the authors only observed 28 natural events of ISs localized into CRISPR-Cas arrays, which may be a result of random integration into bacterial chromosomes. The authors have not provided sufficient evidence to demonstrate that this interaction occurs intentionally, resulting in mutual benefits for both elements. For example, it is unclear whether the CRISPR-Cas loci are disrupted more frequently than other genes, which could suggest a positive association between these elements.

Additionally, while the authors have analyzed the impact of this association using plasmids, it is important to consider the role of CRISPR-Cas systems in blocking phage infections. A strain carrying an inactive CRISPR-Cas system may be eliminated from the population unless the insertion of the ISs can be reverted. Have the authors tested this?

Furthermore, the experiments analyzing fitness costs are questionable. These experiments should ideally be performed in the original species/strain (*E. amylovora*) carrying the system, rather than using the surrogate *E. coli*, as the *E. coli* system is completely artificial. Other option is to use CRISPR-Cas system already present in *E. coli*.

In summary, the results suggest that in a very small subset of strains, the ISs may disrupt the activity of the CRISPR-Cas system. However, based on the low number of events observed, it is difficult to imagine that this interaction is relevant in nature. The authors have not demonstrated a positive association between these elements either.

Reviewer #2 (Remarks to the Author):

The authors have taken a novel and interesting approach to examining the question of whether IS are able to counter CRISPR activity, by interrupting various cas genes, and by doing so are able to drive host evolution?

The authors demonstrate a sound broad understanding of IS transposition mechanisms. The manuscript is well-written, and the experimental aspects of the work are carefully planned and executed. In particular, the amount of work undertaken to capture and analyse the IS transposition events into the experimental cas capture system is commendable.

This is an excellent body of work which, for the first time on a broad scale, examines the interaction between IS and cas clusters.

Comments:

Line 72 The use of the word "adaptable" suggests some sort of active process. In the case of IS1 and IS10, they do not have a known target site, so there is no requirement for them to "adapt" to the sequence evolution of cas. Consider modifying this sentence.

Line 82. The collection of known IS elements should be listed or somehow identified. There are over 4000 IS in the ISFinder database, were all of these used in this analysis?

Line 85. Care needs to be taken when using TSDs as a parameter for analysis. Quite a few IS do not generate TSDs, and for those that do generate TSDs they can be lost if there has been a

subsequent adjacent deletion (commonly seen with IS26 and IS1).

Lines 88-89 and Figure S1. The most striking thing here is the relative lack of diversity in the IS found in the cas clusters. Given how many IS are known to exist, and how many complete genomes have now been reported, I would have expected to see many more different IS, just by chance, interrupting cas clusters. Is there some sort of bias in the CRISPRCasbd? Are there only single entries for each cluster, or are there multiple entries?

Line 285-288. Care needs to be taken with this interpretation. E. coli tend to have large numbers of IS1 copies (as shown by Figure S2), so there is a potential bias built into this assay system. If the strain starts with a higher number of copies of a particular IS, there is naturally a greater chance of those IS being represented in cas-disrupting events than the less common IS. Some IS are also inherently more active than others (i.e. IS1 and IS10 have naturally higher transposition frequencies than IS5 which rarely seems to move).

Reviewer #3 (Remarks to the Author):

The authors identify a CRISPR system carrying an IS10 insertion in *Erwinia amylovora*. They then express this system heterologously in an *E. coli* host, along with an additional plasmid that carries protospacers targeted by the CRISPR system. They demonstrate that such targeting is deleterious in the presence of CRISPR, but that an inactivated CRISPR system removes this cost. Such inactivation can also lead to a positive interaction between host and plasmid when the plasmid carries antibiotic resistance markers and antibiotic selection is applied. This element of the paper is convincing but perhaps somewhat unsurprising.

They then design an assay to identify IS insertions into cas genes in a high throughput way, by adding a self-targeting guide coupled with an inducible Cas9 system. This allows them to probe what determines insertion into cas genes and identify the relative frequencies of different IS elements.

Lastly, they use a bioinformatic screen to assess the distributions of IS elements across taxa and within specific anti-MGE defences, such as RM etc.

The experiments are comprehensive and the bioinformatic analysis seems valuable for describing the distributions of various IS elements. My main concerns are that this may not be of particular relevance to anyone other than those studying IS elements and that the narrative of the paper is not clear. I am also not really persuaded that the evidence presented demonstrates that IS elements are 'activated' (line 197) to insert into cas genes (versus selection after a random event).

Some of the results are very interesting, such as the observation that IS insertion into cas genes much more common than SNPs and indels- potentially speeding up the evolutionary response during selection. I'm also not very familiar with the IS literature, and some of these observations may be quite novel i.e. which IS elements are most prevalent in *E. coli*, and which have the ability to tolerate a range of insertion sites / motifs. Perhaps with a clearer narrative this paper would appeal to a wider audience.

In its present form I found it hard to follow the narrative and rationale for some of the experiments, and it was unclear to me why this would be of interest to those beyond the immediate IS field.

Minor:

Line 37- is CRISPR really more dominant than RM systems? Probably not given prevalence of both

systems.

Line 68: "we hypothesized that the collapse of CRISPR machinery might be a fitness cost of the host during stress survival". This is unclear to me. Do the authors mean that the loss of CRISPR is the fitness cost- which is possible if susceptibility to parasitic MGEs increases susceptibility to phages, for example? I could see a scenario where, under stress, the host needs to adapt, and susceptibility to incoming plasmids could be highly beneficial if they confer advantageous traits. Some clarity around this would help.

Line 121: SYH01 and SYH02. Is the only difference between these hosts the vector (pEraCas or pEraCas-IS10)? If so, why not simply refer to the plasmid names?

Line 242- regarding whether the insertion is random or specific to cas genes. Can the authors be sure that there is no cost to carrying the CRISPR-expressing plasmid, even in the absence of a targeting guide? If there is an associated cost to carriage (metabolic / autoimmune growth defect etc) then there is still likely to be selection for inactivation of the cas gene. I'm not sure this result alone is sufficient to state that IS integration into the cas gene is non-random. Showing the background rate of insertion across the genome, and completely ruling out selection, would be required for this.

Line 398: Is it possible that E.coli having the richest diversity of IS elements simply reflects the biases in the database used and they are of no more (or less) benefit than in other hosts?

Fig 2C: Is it necessary to show the 3 concentrations of antibiotics used when the result is almost identical?

Fig. 3. Surprising that the self-targeting mutant, with all p7-3 protospacers integrated into the chromosome, can survive so well. Suggests Cas expression very low?

Point-by-point Response to Referees

We sincerely appreciate the reviewers for their insightful comments and constructive suggestions, which have significantly improved the quality of our manuscript. We provide a point-by-point response to all comments from reviewers in **BLUE** font below (the line numbers, figure numbers and table numbers refer to the revised manuscript).

REVIEWER COMMENTS

Reviewer #1 (Remarks to the Author):

A fascinating and rapidly advancing area of study in bacteriology is focused on unraveling how various mobile genetic elements collaborate to drive bacterial evolution. In this manuscript, the authors delve into the intricate interplay between insertion sequences (ISs) and the CRISPR-Cas system. Their proposed hypothesis suggests that ISs can potentially disrupt the activity of the CRISPR-Cas system, thereby increasing the ability of recipient cells to acquire foreign DNA.

Response: We sincerely appreciate the reviewer's commendation of our work and the insightful suggestions provided. We have addressed all the comments listed below with additional experiments, bioinformatic analyses and have revised the manuscript accordingly. We hope these revisions have significantly improved our manuscript.

While this idea is interesting, this reviewer feels that the manuscript is in its early stages, and additional experiments are required to support this hypothesis. Specifically, the authors only observed 28 natural events of ISs localized into CRISPR-Cas arrays, which may be a result of random integration into bacterial chromosomes. The authors have not provided sufficient evidence to demonstrate that this interaction occurs intentionally, resulting in mutual benefits for both elements. For example, it is unclear whether the CRISPR-Cas loci are disrupted more frequently than other genes, which could suggest

a positive association between these elements.

Response: As described in Methods section, we identified 28 cases of ISs transpositions into *cas* genes by performing a stringent BLASTn search, aligning ISs from a manually curated ISs database (ISfinder) to putative CRISPR/Cas gene clusters obtained from the CRISPRCasdb database, with 100% sequence identity and coverage cut-off. We initially adopted an extremely stringent cut-off and the manually curated database to ensure higher analytical accuracy. However, since the ISs in the ISfinder repository encompass only a fraction of the ISs present in the public databases¹, it is conceivable that our analysis has only revealed a portion of the actual transposition events.

Supplementary information, Figure S2. Identification of naturally occurring transpositions of ISs into *cas* genes using a highly sensitive software pipeline, ISEScan. A total of 163 transposition events are shown above. Each IS element in the

figures represents a unique IS family. Horizontal axes indicate the relative position of each gene or IS within the targeted genomic region.

To overcome the inherent limitations of the database, we further employed a more sensitive software pipeline, ISEScan², which was based on profile hidden Markov models (HMMs) constructed from manually curated ISs, thereby enabling a more comprehensive detection of ISs transpositions. As expected, we identified a total of 163 natural ISs transpositions into *cas* genes within the same CRISPRCasdb database (Figure S2), demonstrating the involvement of 20 different IS families (Figure S3).

Supplementary information, Figure S3. Analysis of ISs engagement in 163 cases of naturally occurring transpositions of ISs into *cas* genes. It is important to note that certain transposition events within the analyzed cases involve the participation of multiple IS families, rather than a single IS family.

Additionally, we also systematically explored the transposition events of the

miniature inverted repeat transposable elements (MITEs)³, which are the non-autonomous IS derivatives and can be potentially catalyzed in trans by the transposase of a related complete IS, within the *cas* genes. We then analyzed the presence of MITEs within *cas* genes of the CRISPRCasdb database using MITE-Tracker pipeline⁴, revealing a similar transposition phenomenon observed with ISs (Figure S4).

Supplementary information, Figure S4. Naturally occurring transpositions of MITEs into *cas* genes. Horizontal axes indicate the relative position of each gene or MITE within the targeted genomic region.

It is also worth noting that CRISPRCasdb database, despite being a highly refined repository of CRISPR/Cas systems, only contains systems derived from complete genomes available in the GenBank database⁵, which may result in the omission of a substantial number of CRISPR/Cas systems, and an underestimation of the actual prevalence of ISs transpositions into *cas* genes. Taken together, given the continual expansion of relevant databases and advancements in bioinformatic tools, we anticipate that a growing number of such transposition events will be detected.

Meanwhile, as the reviewer pointed out, the occurrence of IS transpositions into *cas* genes relative to other genes is yet to be determined. To address this issue, it is imperative to select a suitable comparator with comparable occurrence frequencies to *cas* genes, thereby ensuring the reliability and validity of the comparative analysis. Considering the essential functional synergy between *cas* genes and CRISPR arrays in the proper functioning of the CRISPR/Cas system, wherein CRISPR arrays often co-occur with *cas* genes, we conducted a comparative analysis to assess the frequency of IS transpositions into the two elements within the CRISPRCasdb database. Remarkably, our analysis revealed a clear propensity for ISs transpositions, with exclusive occurrences observed in *cas* genes, while no IS insertions were identified within the CRISPR arrays, indicating a non-random process of ISs integrations into *cas* genes.

Overall, we think that our revisions (line 108) suggested by the reviewer have significantly improved our study, particularly in elucidating the natural occurrence of ISs transpositions into *cas* genes.

Additionally, while the authors have analyzed the impact of this association using plasmids, it is important to consider the role of CRISPR-Cas systems in blocking phage infections. A strain carrying an inactive CRISPR-Cas system may be eliminated from the population unless the insertion of the ISs can be reverted. Have the authors tested this?

Response: The reviewer raised a very good point. The acquisition of mobile genetic elements (MGEs) through horizontal gene transfer (HGT) is a primary driver of

bacterial evolution. However, the CRISPR/Cas system is a double-edged sword, acting as an adaptive immune defense system that maintain genome stability and integrity by preventing the invasion of foreign genetic material, while simultaneously hindering the acquisition of beneficial genes through HGT. In our study, the deliberate inactivation of the CRISPR/Cas system through ISs transpositions emerges as a strategy enabling rapid acquisition of plasmids, thereby enhancing the survival capabilities of the host under antibiotic stress. The disruption of CRISPR/Cas system indeed leads to a diminished capacity to impede phage infections.

However, it is important to note that the presence of other defense systems, such as restriction modification (RM) system⁶, cyclic-oligonucleotide-based anti-phage signaling system (CBASS)⁷, within the host, can also offer a degree of protection against phage infections. Within their ecological niches, microorganisms encounter diverse environmental pressures (various antibiotic stress, phage infections, *etc.*), and may often be forced to prioritize and make strategic trade-offs in their adaptive responses, aiming to ensure their survival and successful adaptation to prevailing challenges. Below we explore this argument in more detail.

As described in our study, *E. amylovora* 7-3, a globally distributed agricultural pathogen, poses significant economic threats. In the USA and Canada, where *E. amylovora* infections are prevalent, and the control of this pathogen commonly relies on the targeted application of the antibiotic streptomycin⁸. Through the analysis of spacer sequences within the CRISPR array of *E. amylovora* 7-3, we have identified it possesses an intrinsic type I-E CRISPR/Cas system capable of targeting the plasmid 7-3 that contains streptomycin resistance genes (*strA* and *strB*). When confronted with the predominant challenge of streptomycin treatment, the presence of this antibiotic imposes the most substantial selective pressure on the survival of *E. amylovora* 7-3 compared to other environmental pressures. However, at this juncture, the intrinsic CRISPR/Cas system, which defends against p7-3 plasmid invasion, proves detrimental to the host. This antibiotic selective pressure acts as a strong driving force for the inactivation of the strain's CRISPR/Cas system through IS10 transposition, thereby facilitating its coexistence with the p7-3 plasmid. Therefore, this coexistence enables

the acquisition of streptomycin resistance genes encoded on the p7-3 plasmid, allowing *E. amylovora* 7-3 to persist and adapt in the face of streptomycin treatment. We hope that this example provides further insights into the biological significance of ISs transpositions into *cas* genes.

About the point 2 (a strain carrying an inactive CRISPR-Cas system may be eliminated from the population unless the insertion of the ISs can be reverted. Have the authors tested this?) raised by the reviewer, we have reviewed the relevant literature in an effort to address this concern. ISs can propagate in the genome via copy-and-paste or cut-and-paste transposition mechanisms. In the copy-and-paste mechanism, IS replicates itself and inserts a duplicated copy into a different genomic location, while preserving its original site. Conversely, the cut-and-paste mechanism involves excising the IS from its original position and relocating it to a new genomic site. Accordingly, for ISs that employ a copy-and-paste transposition mechanism, even if the ISs within the *cas* genes undergo further transposition, they would still remain in their original positions, rendering the Cas proteins inactive. In the case of ISs utilizing a cut-and-paste transposition mechanism, although the ISs indeed are excised from their original positions and transpose away upon subsequent transposition, the target site duplications (TSDs), generated during the initial transposition, would still persist within the *cas* genes and pose a significant risk of disrupting the proper functioning of the CRISPR/Cas system. Consequently, the transpositions of ISs into *cas* genes may result in irreversible disruption. Nevertheless, as previously mentioned, the host possesses other alternative defense systems that act as safeguards, thereby mitigating the risk of a complete breakdown in the defense mechanism.

Furthermore, the experiments analyzing fitness costs are questionable. These experiments should ideally be performed in the original species/strain (*E. amylovora*) carrying the system, rather than using the surrogate *E. coli*, as the *E. coli* system is completely artificial. Other option is to use CRISPR-Cas system already present in *E. coli*.

Response: As the reviewer suggested, we utilized the endogenous type I-E CRISPR/Cas system of *E. coli* DH10B to further explore the role of ISs in balancing genetic defense and the benefits of plasmid acquisition. *E. coli* DH10B is equipped with a type I-E CRISPR/Cas system comprising eight *cas* genes (*cas1*, *cas2*, *cas3* and *casABCDE*) and two CRISPR arrays (CRISPR I and CRISPR II)⁹. These eight *cas* genes were reported to be organized in two operons (*casA-casB-casC-casD-casE-cas1-cas2* and *cas3*). Under normal conditions, the type I-E CRISPR/Cas system of *E. coli* DH10B remains silent due to strong repression of *casABCDE12* operon by the histone-like nucleoid structuring protein (H-NS)^{10,11}. In light of this, we replaced the native promoters of *casABCDE12* operon and *cas3* with a constitutive promoter J23119 and an arabinose-inducible promoter (pBAD), respectively, generating strain ActSY01 (Figure S5A). Additionally, we designed and inserted two different CRISPR spacers, specifically targeting the protospacers immediately downstream of the AAG (PAM) sequence within both an essential gene (*ftsA*) and a non-essential gene (*arpA*), into a p15A-derived plasmid, generating p15A-CRISPR::*ftsA* and p15A-CRISPR::*arpA* plasmids, respectively. Compared to the control plasmid p15A-sgRNA::*lacZ02* lacking the targeting spacer for the type I-E CRISPR/Cas system, the transformation efficiencies of p15A-CRISPR::*ftsA* and p15A-CRISPR::*arpA* into the strain ActSY01 were dramatically decreased by about four orders of magnitude in the presence of L-arabinose induction (Figure S5B).

Supplementary information, Figure S5. ISs mediate the fitness trade-off between the benefits of acquiring plasmids and the genotoxicity induced by the type I-E CRISPR/Cas system of *E. coli* DH10B under antibiotic pressure. (A) Schematic of the engineered type I-E CRISPR-Cas locus of *E. coli* DH10B. The strain ActSY01 was generated by replacing the native promoters of *casABCDE12* operon and *cas3* with a constitutive promoter J23119 and an arabinose-inducible promoter (pBAD), respectively. Primer pairs P1 and P2 were designed to amplify the entire *cas* cluster of the strain ActSY01; expected amplicon sizes are shown. (B) In the presence of L-arabinose induction, transformation efficiencies of p15A-sgRNA::lacZ02 (control, lacking the targeting spacer for the type I-E CRISPR/Cas system), p15A-CRISPR::arpA (targeting the non-essential gene *arpA* of the strain ActSY01) and p15A-CRISPR::ftsA (targeting the essential gene *ftsA* of the strain ActSY01) into the strain ActSY01 were determined through counting CFUs per 200 ng of plasmids. Error bars denote means ± S.D. from three biological replicates; ****, P < 0.0001, using two-tailed unpaired t-test. (C) Colony PCR screening with primer pairs P1 and P2 for IS insertions into the *cas* cluster of the 23 escapers (A01-A23). Lanes “WT”, control amplicons with the strain ActSY01 template; larger PCR products suggest IS transposition events. Except for

samples A8, A15, A21 and A22, IS insertions were detected within the *cas* genes of all other 19 samples. Detailed information regarding the IS types involved in transposition and their resulting TSD sequences can be found in Table S6.

Nevertheless, a large number of escapers, capable of evading CRISPR/Cas-mediated cleavage, emerged under the selective pressure of chloramphenicol antibiotic. To further explore the underlying immune escape mechanisms, we randomly selected 23 escapers from the strains ActSY01 harboring p15A-CRISPR::ftsA or p15A-CRISPR::arpA, and subsequently performed PCR amplification and Sanger sequencing of the complete chromosomal *cas* operons. As expected, we detected various compensatory mutations corresponding to the transpositions of IS1, IS2, IS5 and IS10 into the *cas* genes within the intrinsic type I-E CRISPR/Cas system of the strain ActSY01 (Figure S5C and Table S6).

Supplementary information, Table S6. Analysis of a total of 19 larger amplicons containing an IS element insertion into *cas* genes within the type I-E CRISPR/Cas system of the strain ActSY01.

Samples	IS type	TSD sequences	TSD length
A01	IS10	CGCAGAACA	9
A02	IS10	TGCTATGCT	9
A03	IS10	TGTCAGGCA	9
A04	IS10	TGCTATGCT	9
A05	IS5	TTAG	4
A06	IS1	ACGCCAGCG	9
A07	IS1	GGCAATGGC	9
A09	IS2	GGCAC	5
A10	IS10	TGCTATGCT	9
A11	IS10	GGCCTGGCT	9
A12	IS10	CACTCGACG	9
A13	IS5	TTAA	4
A14	IS10	CACTCGACG	9
A16	IS10	TGCTGCGCA	9
A17	IS10	CACTCGACG	9
A18	IS1	CTGGTTTCC	9
A19	IS10	GGCCTGGCT	9
A20	IS10	AGCTTCGCC	9
A23	IS10	CGCAGAACA	9

These findings indicated that through ISs-mediated disruption of their native CRISPR/Cas system, host achieved coexistence with plasmids that targeted the chromosome, effectively eliminating the CRISPR/Cas-induced genotoxicity while simultaneously acquiring antibiotic resistance gene encoded by the plasmids, enabling survival under antibiotic pressure. Remarkably, these results further supported our studies on the type I-E CRISPR/Cas system of *E. amylovora* (Figure 3), the type II CRISPR/Cas9 system of *Streptococcus pyogenes*, and the type V CRISPR/Cpf1 system of *Francisella novicida* (Figure 5).

On the other hand, in response to the reviewer's comment regarding the “artificial system”, we apologize for any lack of clarity in our manuscript. We would like to provide additional information to address this concern. To streamline the characterization of ISs transpositions and establish an efficient ISs-trapping system, we utilized type II (SpCas9) and type V (FnCpf1) CRISPR/Cas systems with a single multidomain effector protein arbitrating both recognition and cleavage in *E. coli*. Such surrogate *E. coli*-based artificial CRISPR interference system presented a notable advantage over the complex type I-E CRISPR/Cas system, as it only required the detection of a single gene during ISs transpositions analysis, while the latter necessitated the simultaneous identification of eight genes. Moreover, the reconstruction of nucleotide sequences on a single effector protein was notably easier compared to eight proteins, thus greatly facilitating our investigation into the responses of ISs to the rapid and forced evolution of Cas variants (Figure 5).

In summary, the results suggest that in a very small subset of strains, the ISs may disrupt the activity of the CRISPR-Cas system. However, based on the low number of events observed, it is difficult to imagine that this interaction is relevant in nature. The authors have not demonstrated a positive association between these elements either.

Response: Following the reviewer’s valuable suggestions, we employed the highly sensitive ISs detection tool within the same CRISPRCasdb database and identified a total of 163 natural ISs transpositions into *cas* genes, involving the participation of 20

distinct IS families and spanning across 57 different genera (Table S5). Furthermore, we also explored the biological significance of ISs transpositions into *cas* genes using the endogenous type I-E CRISPR/Cas system of *E. coli* DH10B. Detailed insights regarding these results are provided in the preceding responses.

Supplementary information, Table S5. Taxonomic analysis of 163 cases of ISs transpositions into *cas* genes.

accession number	taxid	species	genus
AP024271.1	274	Thermus thermophilus	Thermus
CP039845.1	438	Acetobacter pasteurianus	Acetobacter
LR134176.1	446	Legionella pneumophila	Legionella
CP023258.1	562	Escherichia coli	Escherichia
CP027388.1	562	Escherichia coli	Escherichia
CP034843.1	562	Escherichia coli	Escherichia
AP019189.1	562	Escherichia coli	Escherichia
CP051716.1	562	Escherichia coli	Escherichia
CP055669.1	562	Escherichia coli	Escherichia
CP024260.1	562	Escherichia coli	Escherichia
CP043739.1	562	Escherichia coli	Escherichia
CP047571.1	562	Escherichia coli	Escherichia
CP007265.1	562	Escherichia coli	Escherichia
CP018995.1	562	Escherichia coli	Escherichia
CP025851.1	562	Escherichia coli	Escherichia
CP017844.1	562	Escherichia coli	Escherichia
CP025840.1	562	Escherichia coli	Escherichia
CP007391.1	562	Escherichia coli	Escherichia
CP027597.1	562	Escherichia coli	Escherichia
CP044315.2	562	Escherichia coli	Escherichia
CP014197.1	562	Escherichia coli	Escherichia
CP044346.1	562	Escherichia coli	Escherichia
CP057838.1	562	Escherichia coli	Escherichia
CP059288.1	562	Escherichia coli	Escherichia
CP029981.1	562	Escherichia coli	Escherichia
LT883142.1	562	Escherichia coli	Escherichia
CP042606.1	562	Escherichia coli	Escherichia
CP043750.1	562	Escherichia coli	Escherichia
CP027319.1	562	Escherichia coli	Escherichia
CP007390.1	562	Escherichia coli	Escherichia
CP041628.1	562	Escherichia coli	Escherichia
CP010143.1	562	Escherichia coli	Escherichia
CP057281.1	562	Escherichia coli	Escherichia

CP056842.1	562	Escherichia coli	Escherichia
CP019259.1	562	Escherichia coli	Escherichia
LR882973.1	562	Escherichia coli	Escherichia
CP027763.1	562	Escherichia coli	Escherichia
CP024659.1	562	Escherichia coli	Escherichia
LR883050.1	562	Escherichia coli	Escherichia
CP057207.1	562	Escherichia coli	Escherichia
CP027140.1	562	Escherichia coli	Escherichia
CP019256.1	562	Escherichia coli	Escherichia
CP022279.1	562	Escherichia coli	Escherichia
CP007594.1	562	Escherichia coli	Escherichia
CP043742.1	562	Escherichia coli	Escherichia
CP010152.1	562	Escherichia coli	Escherichia
LR134236.1	562	Escherichia coli	Escherichia
CP057287.1	562	Escherichia coli	Escherichia
AP021894.1	562	Escherichia coli	Escherichia
CP059929.1	562	Escherichia coli	Escherichia
CP018109.1	562	Escherichia coli	Escherichia
CP057860.1	562	Escherichia coli	Escherichia
CP029687.1	562	Escherichia coli	Escherichia
CP057204.1	562	Escherichia coli	Escherichia
CP057173.1	562	Escherichia coli	Escherichia
CP055945.1	562	Escherichia coli	Escherichia
CP053720.1	562	Escherichia coli	Escherichia
CP056495.1	571	Klebsiella oxytoca	Klebsiella
CP012753.1	573	Klebsiella pneumoniae	Klebsiella
CP026846.1	621	Shigella boydii	Shigella
CP026834.1	622	Shigella dysenteriae	Shigella
CP034935.1	622	Shigella dysenteriae	Shigella
CP026831.1	622	Shigella dysenteriae	Shigella
CP026840.1	622	Shigella dysenteriae	Shigella
CP028249.1	1254	Pediococcus acidilactici	Pediococcus
CP028247.1	1254	Pediococcus acidilactici	Pediococcus
CP012911.1	1307	Streptococcus suis	Streptococcus
CP007587.1	1346	Streptococcus iniae	Streptococcus
CP007586.1	1346	Streptococcus iniae	Streptococcus
CP043615.1	1581	Lentilactobacillus buchneri	Lentilactobacillus
CP041944.1	1597	Lactocaseibacillus paracasei	Lactocaseibacillus
CP034099.1	1613	Limosilactobacillus fermentum	Limosilactobacillus
CP016803.1	1613	Limosilactobacillus fermentum	Limosilactobacillus
CP067088.1	1613	Limosilactobacillus fermentum	Limosilactobacillus
LT618776.1	1744	Propionibacterium freudenreichii	Propionibacterium
LT618777.1	1744	Propionibacterium freudenreichii	Propionibacterium
LR699570.1	1765	Mycobacterium tuberculosis	Mycobacterium

CP041806.1	1773	Mycobacterium tuberculosis	Mycobacterium
CP044345.1	1773	Mycobacterium tuberculosis	Mycobacterium
CP041875.1	1773	Mycobacterium tuberculosis	Mycobacterium
CP027234.1	28113	Bacteroides heparinolyticus	Bacteroides
CP028370.1	28183	Leptospira santarosai	Leptospira
CP034309.1	28200	Aliarcobacter skirrowii	Aliarcobacter
CP029250.1	29397	Lactobacillus delbrueckii	Lactobacillus
CP023139.1	29397	Lactobacillus delbrueckii	Lactobacillus
CP018215.1	29397	Lactobacillus delbrueckii	Lactobacillus
CP051101.1	29495	Vibrio navarrensis	Vibrio
CP026503.1	47770	Lactobacillus crispatus	Lactobacillus
CP046311.1	47770	Lactobacillus crispatus	Lactobacillus
CP012890.1	52242	Lactobacillus gallinarum	Lactobacillus
CP038281.1	54291	Raoultella ornithinolytica	Raoultella
CP037427.1	76832	Myroides odoratimimus	Myroides
CP058954.1	97478	Limosilactobacillus mucosae	Limosilactobacillus
AP018824.1	108980	Acinetobacter ursingii	Acinetobacter
AP024237.1	110505	Mycobacterium heckeshornense	Mycobacterium
CP065155.1	113557	Lacticaseibacillus paracasei	Lacticaseibacillus
CP064311.1	113557	Lacticaseibacillus paracasei	Lacticaseibacillus
CP014342.1	129338	Geobacillus subterraneus	Geobacillus
CP002552.1	153948	Nitrosomonas sp. AL212	Nitrosomonas
CP016786.1	182773	Clostridium isatidis	Clostridium
CP044117.1	207340	Roseomonas mucosa	Roseomonas
CP009512.1	213585	Methanosarcina mazei	Methanosarcina
AE006641.1	273057	Saccharolobus solfataricus	Saccharolobus
AE008691.1	273068	Caldanaerobacter subterraneus	Caldanaerobacter
FN649414.1	316401	Escherichia coli	Escherichia
CP001101.1	331678	Chlorobium phaeobacteroides	Chlorobium
AP008937.1	334390	Limosilactobacillus fermentum	Limosilactobacillus
CP011410.1	338215	Leptospira interrogans	Leptospira
CP003056.1	345219	Weizmannia coagulans	Weizmannia
CP000679.1	351627	Caldicellulosiruptor saccharolyticus	Caldicellulosiruptor
CP001702.1	395962	Rippkaea orientalis	Rippkaea
CP000943.1	426117	Methylobacterium sp. 4-46	Methylobacterium
CM001022.1	584708	Aminomonas paucivorans	Aminomonas
AP012340.1	652616	Mycobacterium tuberculosis	Mycobacterium
CP002432.1	653733	Desulfurispirillum indicum	Desulfurispirillum
CP002539.1	693977	Deinococcus proteolyticus	Deinococcus
CP023669.1	702114	Methylomonas koyamae	Methylomonas
CP002033.1	712938	Limosilactobacillus fermentum	Limosilactobacillus
CP002858.1	717231	Flexistipes sinusarabici	Flexistipes
CM001225.1	882098	Mycobacterium tuberculosis	Mycobacterium
CM001226.1	882099	Mycobacterium tuberculosis	Mycobacterium

CM001227.1	882100	Mycobacterium tuberculosis	Mycobacterium
CP002411.1	929506	Clostridium botulinum	Clostridium unclassified
CP032507.1	947515	Ectothiorhodospiraceae bacterium BW-2	Ectothiorhodospiraceae genus
CP031345.1	1055538	Escherichia coli	Escherichia
CP031349.1	1055538	Escherichia coli	Escherichia
CP031347.1	1055538	Escherichia coli	Escherichia
CP031343.1	1055538	Escherichia coli	Escherichia
CP028379.1	1055544	Escherichia coli	Escherichia
CP031916.1	1078032	Escherichia coli	Escherichia
AP019703.1	1078034	Escherichia coli	Escherichia
CP009709.1	1121088	Weizmannia coagulans	Weizmannia
CP042169.1	1179670	Micrococcus sp. KBS0714	Micrococcus
CP003988.1	1262452	Streptomyces sp. 769	Streptomyces unclassified
AP013042.1	1303921	Bathymodiolus septemdierum thioautotrophic gill symbiont	Gammaproteobacteria genus
HF952106.1	1318615	Streptococcus agalactiae	Streptococcus
CP005941.1	1318633	Streptococcus iniae	Streptococcus
CP011536.1	1381124	Limosilactobacillus fermentum	Limosilactobacillus
AP019551.1	1441386	Athalassotoga saccharophila	Athalassotoga
AP019755.1	1444684	Nitrosomonas stercoris	Nitrosomonas
CP024266.1	1446746	Escherichia coli	Escherichia
CP022912.1	1446746	Escherichia coli	Escherichia
CP007156.1	1451189	Corynebacterium falsenii	Corynebacterium
CP011271.1	1630693	Gemmata sp. SH-PL17	Gemmata
CP014360.1	1727196	Methylomonas sp. DH-1	Methylomonas
CP029145.1	1850093	Hymenobacter nivis	Hymenobacter unclassified Nostocales
CP019636.1	1940762	Nostocales cyanobacterium HT-58-2	genus
CP032143.1	2004647	Acinetobacter sp. WCHAc010052	Acinetobacter
AP023240.1	2020875	Methyloprofundus sp.	Methyloprofundus
CP042909.1	2047767	Thermosulfurimonas marina	Thermosulfurimonas
CP024955.1	2055160	Kyrpidia spormannii	Kyrpidia
CP035122.1	2505977	Halorussus sp. RC-68	Halorussus
CP043217.1	2605620	Escherichia coli	Escherichia
CP044222.1	2614693	Nitrincola iocasae	Nitrincola
CP048030.1	2698684	Sinimariniibacterium sp. NLF-5-8	Sinimariniibacterium
AP023239.1	2741301	Paenibacillus sp. URB8-2	Paenibacillus
AP021879.1	2752305	Desulfosarcina ovata	Desulfosarcina
CP059399.1	2755382	Nocardia huaxiensis	Nocardia
CP065383.1	2847778	Atribacter laminatus	Atribacter

Again, we deeply appreciate the reviewer's insightful comments, which have

greatly enhanced the overall quality and clarity of our work.

Reviewer #2 (Remarks to the Author):

The authors have taken a novel and interesting approach to examining the question of whether IS are able to counter CRISPR activity, by interrupting various *cas* genes, and by doing so are able to drive host evolution?

The authors demonstrate a sound broad understanding of IS transposition mechanisms. The manuscript is well-written, and the experimental aspects of the work are carefully planned and executed. In particular, the amount of work undertaken to capture and analyse the IS transposition events into the experimental *cas* capture system is commendable.

This is an excellent body of work which, for the first time on a broad scale, examines the interaction between IS and *cas* clusters.

Response: We sincerely appreciate the reviewer's high evaluation of our work, which reinforces the significance and validity of our findings.

Comments:

Line 72 The use of the word “adaptable” suggests some sort of active process. In the case of IS1 and IS10, they do not have a known target site, so there is no requirement for them to “adapt” to the sequence evolution of *cas*. Consider modifying this sentence.

Response: We appreciate the reviewer's valuable suggestions, and we have revised this sentence accordingly. (line 74)

Line 82. The collection of known IS elements should be listed or somehow identified. There are over 4000 IS in the ISFinder database, were all of these used in this analysis?

Response: We appreciate the reviewer's valuable suggestions. In this study, we utilized the entire collection of ISs curated from the ISfinder database, as described in the Methods section (line 812). Following the reviewer's suggestions, we have further revised this sentence and provided additional details to enhance clarity (line 85).

Line 85. Care needs to be taken when using TSDs as a parameter for analysis. Quite a

few IS do not generate TSDs, and for those that do generate TSDs they can be lost if there has been a subsequent adjacent deletion (commonly seen with IS26 and IS1).

Response: The reviewer raised a very good point. Indeed, as pointed out by the reviewer, certain ISs do not generate TSDs during transpositions. Given that the presence of TSDs could serve as evidence of transposition events, we initially applied a stringent analysis approach to identify ISs transpositions into *cas* genes. While this conservative method may underestimate the true number of transposition events, it ensures the validity and the reliability of the results.

In our initial analysis, as described in the Methods section, we identified 28 cases of ISs transpositions into *cas* genes by performing a stringent BLASTn search, aligning ISs from a manually curated ISs database (ISfinder) to putative CRISPR-Cas gene clusters obtained from the CRISPRCasdb database, with 100% sequence identity and coverage cut-off. We initially adopted an extremely stringent cut-off and the manually curated database to ensure higher analytical accuracy. However, since the ISs in the ISfinder repository encompass only a fraction of the ISs present in the public databases¹, it is conceivable that our analysis has only revealed a portion of the actual transposition events. To overcome this limitation and efficiently address the reviewer's concerns, we further employed a highly sensitive software pipeline, ISEScan², which was based on profile hidden Markov models (HMMs) constructed from manually curated ISs, thereby enabling a more comprehensive detection of ISs transpositions. As expected, we identified a total of 163 natural ISs transpositions into *cas* genes within the same CRISPRCasdb database (Figure S2), demonstrating the involvement of 20 different IS families (Figure S3).

Supplementary information, Figure S2. Identification of naturally occurring transpositions of ISs into *cas* genes using a highly sensitive software pipeline, ISEScan. A total of 163 transposition events are shown above. Each IS element in the

figures represents a unique IS family. Horizontal axes indicate the relative position of each gene or IS within the targeted genomic region.

Supplementary information, Figure S3. Analysis of ISs engagement in 163 cases of naturally occurring transpositions of ISs into *cas* genes. It is important to note that certain transposition events within the analyzed cases involve the participation of multiple IS families, rather than a single IS family.

Additionally, we also systematically explored the transposition events of the miniature inverted repeat transposable elements (MITEs)³, which are the non-autonomous IS derivatives and can be potentially catalyzed in trans by the transposase of a related complete IS, within the *cas* genes. We then analyzed the presence of MITEs within *cas* genes of the CRISPRCasdb database using MITE-Tracker pipeline⁴, revealing a similar transposition phenomenon observed with ISs (Figure S4).

Supplementary information, Figure S4. Naturally occurring transpositions of MITEs into *cas* genes. Horizontal axes indicate the relative position of each gene or MITE within the targeted genomic region.

Lines 88-89 and Figure S1. The most striking thing here is the relative lack of diversity in the IS found in the *cas* clusters. Given how many IS are known to exist, and how many complete genomes have now been reported, I would have expected to see many more different IS, just by chance, interrupting *cas* clusters. Is there some sort of bias in

the CRISPRCasbd? Are there only single entries for each cluster, or are there multiple entries?

Response: The observed limited diversity of ISs within the *cas* clusters may be attributed, in part, to the stringent analytical approach employed in our study. As the responses above, this conservative methodology, while ensuring a high level of accuracy, inherently constrains the range of detected ISs transpositions into *cas* genes. For a more comprehensive analysis of these transposition events, we applied a highly sensitive software pipeline, ISEScan. This pipeline utilizes profile hidden Markov models (HMMs), thus reducing the high dependency on the manually curated ISfinder database and enhancing the detection of ISs transpositions into *cas* genes. As expected, we identified a total of 163 natural ISs transpositions into *cas* genes within the same CRISPRCasdb database (Figure S2), demonstrating the involvement of 20 different IS families (Figure S3). Additionally, we also identified MITEs, being non-autonomous derivatives of ISs, within the *cas* genes (Figure S4). For more details, please refer to the responses above.

On the other hand, it is worth noting that CRISPRCasdb database, despite being a highly refined repository of CRISPR/Cas systems, only contains systems derived from complete genomes available in the GenBank database⁵. This selection criteria of the CRISPRCasdb database may lead to the exclusion of a substantial number of CRISPR/Cas systems that have not been sequenced or are derived from partially assembled contig sequences, thereby potentially resulting in an underestimation of the actual prevalence of ISs transpositions into *cas* genes. Taken together, given the continual expansion of relevant databases and advancements in bioinformatic tools, we anticipate that a growing number of such transposition events will be detected.

Line 285-288. Care needs to be taken with this interpretation. *E. coli* tend to have large numbers of IS1 copies (as shown by Figure S2), so there is a potential bias built into this assay system. If the strain starts with a higher number of copies of a particular IS, there is naturally a greater chance of those IS being represented in *cas*-disrupting events

than the less common IS. Some IS are also inherently more active than others (i.e. IS1 and IS10 have naturally higher transposition frequencies than IS5 which rarely seems to move).

Response: We appreciate the reviewer's valuable suggestions. We have carefully reconsidered and revised the interpretation accordingly (line 352).

Again, we are grateful to the reviewer's insightful suggestions, which have significantly improved the clarity and rigor of our manuscript.

Reviewer #3 (Remarks to the Author):

The authors identify a CRISPR system carrying an IS10 insertion in *Erwinia amylovora*. They then express this system heterologously in an *E. coli* host, along with an additional plasmid that carries protospacers targeted by the CRISPR system. They demonstrate that such targeting is deleterious in the presence of CRISPR, but that an inactivated CRISPR system removes this cost. Such inactivation can also lead to a positive interaction between host and plasmid when the plasmid carries antibiotic resistance markers and antibiotic selection is applied. This element of the paper is convincing but perhaps somewhat unsurprising.

They then design an assay to identify IS insertions into *cas* genes in a high throughput way, by adding a self-targeting guide coupled with an inducible Cas9 system. This allows them to probe what determines insertion into *cas* genes and identify the relative frequencies of different IS elements.

Lastly, they use a bioinformatic screen to assess the distributions of IS elements across taxa and within specific anti-MGE defences, such as RM etc.

The experiments are comprehensive and the bioinformatic analysis seems valuable for describing the distributions of various IS elements. My main concerns are that this may not be of particular relevance to anyone other than those studying IS elements and that the narrative of the paper is not clear.

Response: We sincerely appreciate the reviewer's high evaluation of our work regarding both the experimental and bioinformatic analysis sections. Meanwhile, we apologize for any confusion or insufficient emphasis on the significance of our work in the manuscript. In response, we are committed to addressing these concerns by providing a more thorough discussion regarding the broader implications and significance of our findings.

Despite their vital roles in safeguarding prokaryotes against invasive MGEs, CRISPR/Cas systems may also inadvertently target beneficial foreign DNA. Therefore, it is significant to understand how hosts alleviate the adverse impact of CRISPR/Cas-mediated cleavage when it proves detrimental to hosts. In our study, we revealed that IS elements could mediate the fitness trade-off between the benefits of acquiring MGEs

and genetic defense of their prokaryotic hosts by transposition into *cas* genes. The deliberate inactivation of the CRISPR/Cas system through ISs transpositions emerges as a strategy that facilitates rapid acquisition of plasmids harboring antibiotic resistance genes, thereby enhancing the survival capabilities of the host under antibiotic stress. By employing the highly efficient CRISPR/Cas-mediated IS trapping system, we have successfully characterized a remarkable total of 1457 IS transposition events specifically occurring within various *cas* genes, generating a comprehensive collection of TSD sequences for these ISs. To the best of our knowledge, our study presents the first comprehensive analysis of the interaction between ISs and CRISPR/Cas systems on a broad scale, and uncovers a distinct arsenal for disabling the CRISPR/Cas systems.

Currently, only two primary mechanisms that disrupt the proper functioning of CRISPR/Cas systems have been reported in the literature. MGEs have evolved anti-CRISPR (Acr) proteins which can counteract the immune response to CRISPR/Cas system at three different stages, such as the inhibition of CRISPR/Cas complex stabilization or formation, as well as target DNA recognition or cleavage¹². In addition, temperate prophages typically encode the site-specific integrase capable of catalyzing unidirectional recombination between their *attP* attachment site and *attB* recognition site within the direct repeats of the CRISPR locus, thereby also impairing the functionality of CRISPR/Cas systems¹³. Compared to these two reported anti-CRISPR mechanisms, ISs, with their pervasive presence and broad target recognition capabilities, emerge as a distinct mechanism for interfering with CRISPR/Cas systems. Through unraveling the mechanism and consequences of ISs-mediated CRISPR/Cas disruption, our study provides profound insights into the intricate strategies employed by microorganisms in their perpetual battle against invasive genetic elements, ultimately deepening our understanding of microbial evolution, and shedding light on the broader mechanisms that drive genetic diversity as well as the adaptive responses in microbial communities.

Moreover, our findings have also uncovered the inherent risk posed by chromosome-encoded ISs, as they are capable of impairing the proper functioning of CRISPR/Cas systems, resulting in numerous escapers during genome editing. This

work underscores the need for careful considerations regarding host selection to minimize the impact of ISs-mediated escape events, thereby ensuring the desired genome-editing efficiency. In the case of *E. coli*, a judicious choice would be the IS-free MDS42 strain¹⁴ for gene editing, thereby eliminating any gene editing escape events arising from ISs transpositions into *cas* genes and ensuring successful editing outcomes. Furthermore, the discovery of such anti-CRISPR mechanism highlights the importance of considering not just the off-target effects but also the potential disruptive effects of host-encoded ISs on CRISPR/Cas functionality during gene editing, thereby optimizing gene editing efficiency and accelerating the further applications of CRISPR/Cas systems.

Overall, our study elucidates the significant roles of ISs in mediating the fitness trade-off between the benefits of acquiring MGEs and genetic defense in prokaryotic hosts, and also reveals the impact of ISs on CRISPR/Cas-mediated gene editing efficiency, thus advancing the applications of CRISPR/Cas systems across diverse fields.

I am also not really persuaded that the evidence presented demonstrates that IS elements are ‘activated’ (line 197) to insert into *cas* genes (versus selection after a random event). Response: We greatly appreciate the reviewer's thoughtful consideration regarding the usage of the term “activated”. Actually, it has been documented that transposable elements exhibit more active in response to cellular stress, particularly DNA damage^{15,16}. The previous study revealed that the double-strand breaks (DSBs) on the chromosome of *E. coli* could trigger multifaceted responses involving stress and DNA damage pathways. Based on proteomics analysis, they inferred several pathways, such as small RNA, DNA modification, sequence-specific repressors, were responsible for transposon expression¹⁷. Moreover, UV light has been reported to induce IS10 transposition into *E. coli*¹⁶, highlighting the role of environmental stressors in modulating transposable elements behavior. However, despite the presence of experimental evidence supporting this perspective, it is still a matter of ongoing

debate¹⁸. Therefore, we have revised this sentence to ensure the scientific rigor (line 253).

Some of the results are very interesting, such as the observation that IS insertion into *cas* genes much more common than SNPs and indels- potentially speeding up the evolutionary response during selection. I'm also not very familiar with the IS literature, and some of these observations may be quite novel i.e. which IS elements are most prevalent in *E. coli*, and which have the ability to tolerate a range of insertion sites / motifs. Perhaps with a clearer narrative this paper would appeal to a wider audience.

Response: We appreciate the reviewer's high evaluation of our findings and valuable suggestions. Following the reviewer's suggestions, we have further enhanced the descriptions of these findings regarding ISs in the manuscript and provided additional details as below.

Our analysis revealed eight diverse IS families present on the chromosomes of *E. coli* DH10B and *E. coli* MG1655, with considerable variation in copy numbers (Figure S6). Through utilizing the highly efficient CRISPR/Cas-mediated IS trapping system, we successfully characterized a remarkable total of 1457 IS transposition events specifically occurring within various *cas* genes, with the participation of six different IS types, including IS1, IS2, IS3, IS5, IS10 and IS150 (Figure 5). We further identified the motif sequences of the target site duplications (TSDs) generated by each of the six ISs during transpositions (Figure 5B). To the best of our knowledge, our study provided the most comprehensive collection of TSD sequences for these ISs. Notably, through iterative mutagenesis of IS target sites in *cas* genes, IS1 and IS10 emerged as prominent players in disrupting *cas* genes (Figure 5A), and the emergence of non-canonical TSD motifs indicated that IS1 and IS10 possessed significantly broader target spectrums than originally believed (Figure 6). We have refined these relevant findings in the manuscript to enhance the overall clarity of the work.

In it's present form I found it hard to follow the narrative and rationale for some of the

experiments, and it was unclear to me why this would be of interest to those beyond the immediate IS field.

Response: we sincerely apologize for any confusion or lack of emphasis on the significance of our work in the manuscript. Meanwhile, we appreciate the reviewer's valuable feedback and have carefully considered all provided suggestions. We are committed to addressing these concerns by providing a more comprehensive and in-depth discussion on the broader implications and significance of our findings. For more details, please refer to all responses.

Minor:

Line 37- is CRISPR really more dominant than RM systems? Probably not given prevalence of both systems.

Response: We appreciate the reviewer's valuable suggestions, and we have removed the phrase “the most” from this sentence to enhance scientific rigor (line 37).

Line 68: “we hypothesized that the collapse of CRISPR machinery might be a fitness cost of the host during stress survival”. This is unclear to me. Do the authors mean that the loss of CRISPR is the fitness cost- which is possible if susceptibility to parasitic MGEs increases susceptibility to phages, for example? I could see a scenario where, under stress, the host needs to adapt, and susceptibility to incoming plasmids could be highly beneficial if they confer advantageous traits. Some clarity around this would help.

Response: In deed, the reviewer's understanding aligns with our intentions, meanwhile, we apologize for any confusion resulting from this statement, and we have made the necessary revisions to enhance its clarity (line 69).

Line 121: SYH01 and SYH02. Is the only difference between these hosts the vector (pEraCas or pEraCas-IS10)? If so, why not simply refer to the plasmid names?

Response: Indeed, as accurately described by the reviewer, the strain SYHY01 refers to the *E. coli* DH10B carrying the plasmid pEraCas, while the strain SYHY02

corresponds to the *E. coli* DH10B harboring the plasmid pEraCas-IS10. To simplify the description and avoid lengthy strings during figure annotations or data presentation, we assigned unified numbers to these two strains, which may facilitate the description of transformation efficiencies of other plasmids (e.g., p15A-Eraprotos-spe and p15A-spe in Figure 2B) into these two hosts.

Line 242- regarding whether the insertion is random or specific to *cas* genes. Can the authors be sure that there is no cost to carrying the CRISPR-expressing plasmid, even in the absence of a targeting guide? If there is an associated cost to carriage (metabolic / autoimmune growth defect etc) then there is still likely to be selection for inactivation of the *cas* gene. I'm not sure this result alone is sufficient to state that IS integration into the *cas* gene is non-random. Showing the background rate of insertion across the genome, and completely ruling out selection, would be required for this.

Response: Following the reviewer's valuable suggestions, we further explored the impacts of the expression of SpCas9-HF1 protein with varying cleavage capabilities on the growth of bacterial strains in the absence of sgRNA targeting guides. As described in the Table S2, proteins SpCas9-HF1, SpCas9-HF1 (D10A), SpCas9-HF1 (H840A), and SpCas9-HF1 (D10A&H840A) were individually encoded in DH10B host strains harboring plasmids pCasY-3- Δ SIM-kan, pCasY-3- Δ SIM-kan (D10A), pCasY-3- Δ SIM-kan (H840A), and pCasY-3- Δ SIM-kan (D10A&H840A), respectively. Therefore, we monitored the growth of these four strains under both IPTG-induced and non-induced conditions, and observed that the sole induction of Cas proteins did not result in discernible growth defects or impose significant physiological burdens on the host strains (Figure S8). Additionally, in light of our initial comparative analysis of plasmid transformation efficiencies (Figure 4B), which demonstrated that the presence of significant genotoxicity was observed only when Cas proteins induced double-strand DNA breaks guided by sgRNAs, it could be inferred that the mere expression of Cas proteins is unlikely to promote ISs transpositions into *cas* genes.

Supplementary information, Figure S8. The mere induction of Cas protein expression does not lead to noticeable growth defects in the host strains. Growth curves of the strains encoding different Cas protein variants under both IPTG-induced and non-induced conditions. Shaded areas indicate the means \pm S.D. from five biological replicates.

On the other hand, we apologize for any ambiguity arising from the statement “To determine whether IS targeting was random or specific to the *cas* genes.” In fact, our actual objective was to investigate whether ISs transpositions into *cas* genes could potentially lead to concurrent transpositions into other genomic loci or induce genomic rearrangements and instability. We have revised this sentence for clarity in the manuscript accordingly (line 300).

Line 398: Is it possible that *E. coli* having the richest diversity of IS elements simply reflects the biases in the database used and they are of no more (or less) benefit than in other hosts?

Response: We employed the widely recognized and the most comprehensive ISfinder database¹, a manually curated resource for ISs, and utilized the full NCBI RefSeq database containing the unique genomic sequences of archaea, bacteria and viruses, as our reference genome database. Given these reliable sources, it is unlikely that biases in the database used would lead to this conclusion.

However, we sincerely apologize for the incomplete description of this conclusion in our manuscript. Actually, as shown in Figure S20, *Klebsiella* and *Salmonella* genera, along with *Escherichia* genus, exhibited a high diversity and abundance of ISs. The

remarkable genomic plasticity of these genera, coupled with their tendency to harbor diverse mobile genetic elements, likely contributed to the high diversity and abundance of ISs. These mobile genetic elements, such as plasmids, integrons and transposons, can serve as vehicles for the horizontal transfer of ISs, thereby facilitating their dissemination and proliferation throughout the genomes of these genera. Indeed, the widespread presence of ISs confers particular survival advantages to these genera. Firstly, ISs, with their high mobility and capability to insert into various genomic sites, significantly contribute to genome rearrangements, genetic diversification, and adaptive responses to the new environments by generating genetic variations through transposition events, including insertions, deletions and inversions. Secondly, the composite transposons often consist of two ISs flanking cargo genes, including antibiotic resistance genes, virulence factors and metabolic genes. Consequently, the presence of ISs, particularly within the composite transposons, exerts a profound influence on the horizontal transfer and extensive dissemination of these cargo genes among microbial communities, thereby facilitating the survival and persistence of these genera in diverse ecological niches by conferring antibiotic resistance, enhancing virulence or enabling the utilization of specific nutrients.

We appreciate the reviewer's valuable suggestions, and we have made further improvements to the manuscript to enhance clarity and significance of this conclusion.

Fig 2C: Is it necessary to show the 3 concentrations of antibiotics used when the result is almost identical?

Response: We appreciate the reviewer's valuable suggestions, and we have removed the redundant figures from the manuscript accordingly. The revised figure is shown below.

Figure 2. ISs-deactivated CRISPR-Cas system facilitates the rapid acquisition and improved persistence of the antibiotic resistance plasmid containing identifiable protospacers.

Fig. 3. Surprising that the self-targeting mutant, with all p7-3 protospacers integrated into the chromosome, can survive so well. Suggests Cas expression very low?

Response: Our CRISPR interference assays revealed a slight reduction in the transformation efficiency of pEraCas (with an intact type I-E CRISPR/Cas system of *E. amylovora* 7-3) into SYHY03 compared to that of pEraCas-IS10 containing an IS10-aborted CRISPR/Cas system, however, a substantial number of transformants still emerged (Figure 3C). The results indicated that the CRISPR/Cas system derived from *E. amylovora* 7-3 did not exhibit effective cleavage of the protospacers on SYHY03 chromosome. Given that the native promoter within CRISPR/Cas system of *E. amylovora* 7-3 may not work well in the heterologous *E. coli* host, we further replaced it with the L-arabinose-inducible pBAD promoter, generating the plasmid pEraCas-pBAD. The subsequent RT-qPCR analysis revealed that the transcription levels of *cas* operon driven by the native promoter of *E. amylovora* 7-3, were comparable to the basal levels observed in DH10B carrying pEraCas-pBAD in the absence of L-arabinose, while significantly lower than those observed with L-arabinose induction (Figure 3B).

Moreover, in the absence of L-arabinose, the transformation efficiency of pEraCas-pBAD into SYHY03 was comparable to that of pEraCas, while a significant upregulation of *cas* operons (with L-arabinose induction) resulted in a considerable three-logs reduction in transformation efficiency (Figure 3C). Taken together, these results indicated that the robust survival of the self-targeting mutant SYHY03 carrying the self-targeting plasmid pEraCas, could be attributed to the relatively low expression level of the *cas* operon within pEraCas, which did not induce severe cleavage toxicity on the chromosome.

Again, we are grateful to the reviewer's insightful suggestions, which have significantly improved the clarity and rigor of our manuscript.

References

1. Siguier, P., Perochon, J., Lestrade, L., Mahillon, J. & Chandler, M. ISfinder: the reference centre for bacterial insertion sequences. *Nucleic Acids Res.* **34**, D32-D36 (2006).
2. Xie, Z. & Tang, H. ISEScan: automated identification of insertion sequence elements in prokaryotic genomes. *Bioinformatics.* **33**, 3340-3347 (2017).
3. Guo, Z. *et al.* Miniature inverted-repeat transposable elements drive rapid microRNA diversification in Angiosperms. *Mol Biol Evol.* **39** (2022).
4. Crescente, J.M., Zavallo, D., Helguera, M. & Vanzetti, L.S. MITE Tracker: an accurate approach to identify miniature inverted-repeat transposable elements in large genomes. *BMC Bioinformatics.* **19**, 348 (2018).
5. Pourcel, C. *et al.* CRISPRCasdb a successor of CRISPRdb containing CRISPR arrays and *cas* genes from complete genome sequences, and tools to download and query lists of repeats and spacers. *Nucleic Acids Res.* **48**, D535-D544 (2020).
6. Wilson, G.G. & Murray, N.E. Restriction and modification systems. *Annu Rev Genet.* **25**, 585-627 (1991).
7. Millman, A., Melamed, S., Amitai, G. & Sorek, R. Diversity and classification of cyclic-oligonucleotide-based anti-phage signalling systems. *Nat Microbiol.* **5**, 1608-1615 (2020).
8. Stockwell, V.O. & Duffy, B. Use of antibiotics in plant agriculture. *Rev Sci Tech.* **31**, 199-210 (2012).
9. Brouns, S.J. *et al.* Small CRISPR RNAs guide antiviral defense in prokaryotes. *Science.* **321**, 960-964 (2008).
10. Westra, E.R. *et al.* H-NS-mediated repression of CRISPR-based immunity in *Escherichia coli* K12 can be relieved by the transcription activator LeuO. *Mol Microbiol.* **77**, 1380-1393 (2010).
11. Pul, U. *et al.* Identification and characterization of *E. coli* CRISPR-cas promoters and their silencing by H-NS. *Mol Microbiol.* **75**, 1495-1512 (2010).
12. Pawluk, A., Davidson, A.R. & Maxwell, K.L. Anti-CRISPR: discovery,

- mechanism and function. *Nat Rev Microbiol.* **16**, 12-17 (2018).
13. Varble, A. *et al.* Prophage integration into CRISPR loci enables evasion of antiviral immunity in *Streptococcus pyogenes*. *Nat Microbiol.* **6**, 1516-1525 (2021).
 14. Pósfai, G. *et al.* Emergent properties of reduced-genome *Escherichia coli*. *Science.* **312**, 1044-1046 (2006).
 15. Horváth, V., Merenciano, M. & González, J. Revisiting the relationship between transposable elements and the eukaryotic stress response. *Trends Genet.* **33**, 832-841 (2017).
 16. Eichenbaum, Z. & Livneh, Z. UV light induces IS10 transposition in *Escherichia coli*. *Genetics.* **149**, 1173-1181 (1998).
 17. Fan, C. *et al.* Defensive function of transposable elements in bacteria. *ACS Synth Biol.* **8**, 2141-2151 (2019).
 18. de Souza, F.S., Franchini, L.F. & Rubinstein, M. Exaptation of transposable elements into novel cis-regulatory elements: is the evidence always strong? *Mol Biol Evol.* **30**, 1239-1251 (2013).

Reviewer #1 (Remarks to the Author):

The authors have performed additional experiments and addressed the majority of my previous comments. One potential avenue for further analysis that I suggest is investigating whether the ISs are capable of excising themselves from the bacterial chromosome, thereby restoring the functionality of the CRISPR-Cas array. A straightforward approach to explore this possibility would be to employ PCR techniques.

Reviewer #2 (Remarks to the Author):

The authors have adequately addressed my comments in their revised manuscript.

Reviewer #3 (Remarks to the Author):

I find the study convincing that ISs can mediate the trade-off between CRISPR immunity and the acquisition of beneficial MGEs. This element of the work is quite thorough and the majority of my concerns have been addressed adequately.

I still have one issue in regards to the occurrence of IS transpositions into cas genes relative to other genes, or put another way how intentional or specific these insertions are. The suggested analysis in the rebuttal that compares IS transpositions into cas genes vs. CRISPR arrays is inadequate because these genes are functionally linked and therefore not independent. For example, if an IS element inserts into a cas gene, then the CRISPR system becomes inert, meaning there will be no additional effect from subsequent insertions into the arrays. CRISPR arrays may also be more recombinogenic due to the repeat sequences, which might influence IS stability. Why not compare overall IS rate across the whole genome, or per kb?

With that said, this does not diminish the functional implications of insertions into cas genes. I agree with the overall conclusions that such ISs can balance the costs and benefits of MGEs. Adding this distinction to the discussion would probably be sufficient.

Point-by-point Response to Referees

We sincerely appreciate the reviewers' high evaluation of the improvement of our manuscript during the revision. In addition, we have carefully addressed the comments below and provided a point-by-point response to all comments from reviewers in **BLUE** font.

REVIEWER COMMENTS

Reviewer #1 (Remarks to the Author):

The authors have performed additional experiments and addressed the majority of my previous comments. One potential avenue for further analysis that I suggest is investigating whether the ISs are capable of excising themselves from the bacterial chromosome, thereby restoring the functionality of the CRISPR-Cas array. A straightforward approach to explore this possibility would be to employ PCR techniques.

Response: We appreciate the reviewer's high evaluation of the improvement of our manuscript during the revision. As pointed out by the reviewer, the relocation of ISs to a new genomic site can be inferred by PCR amplification of their flanking regions. Specifically, the reduction in the size of the PCR product implies the movement of ISs from their original site to a different locus.

For the ISs that generate the target site duplication (TSD) sequences¹ during transpositions, it is worth noting that even after they excise themselves from their original site and transpose to a new location, the TSD sequences generated from their previous transposition event are still retained within the *cas* gene. These remnants, typically consisting of several base pairs (2-14 bp), continue to act as scars, potentially disrupting the open reading frame (ORF) of the *cas* gene and impeding its proper translation.

Certainly, some ISs undergo transposition without generating the TSD sequences. In light of this scenario, when these ISs transpose into the *cas* gene and subsequently

undergo self-excision before translocating to another site, the aforementioned issue of the scar (TSD) formation is effectively circumvented. However, owing to the absence of the homology-directed repair (HDR) mechanisms in prokaryotes, the precise and high-fidelity repair and joining of excision sites following ISs self-excision remain unfeasible. Under such circumstances, the host organism typically employs its specialized low-fidelity DNA polymerases^{2,3} to generate mutations surrounding the self-excision sites of ISs, striving to restore the incision. These mutations commonly involve base deletions, insertions or substitutions, ultimately resulting in the disruption of the ORF of the *cas* gene and compromising the proper expression of Cas protein.

Therefore, it is indeed feasible to utilize PCR techniques to discern whether ISs have transposed out of the *cas* gene. However, even if the transposition does occur, it may not necessarily restore the functionality of the CRISPR/Cas system.

Reviewer #2 (Remarks to the Author):

The authors have adequately addressed my comments in their revised manuscript.

Response: We sincerely appreciate the reviewer's kind comments and constructive suggestions.

Reviewer #3 (Remarks to the Author):

I find the study convincing that ISs can mediate the trade-off between CRISPR immunity and the acquisition of beneficial MGEs. This element of the work is quite thorough and the majority of my concerns have been addressed adequately.

Response: We appreciate the reviewer's high evaluation of the improvement of our manuscript during the revision.

I still have one issue in regards to the occurrence of IS transpositions into *cas* genes relative to other genes, or put another way how intentional or specific these insertions

are. The suggested analysis in the rebuttal that compares IS transpositions into *cas* genes vs. CRISPR arrays is inadequate because these genes are functionally linked and therefore not independent. For example, if an IS element inserts into a *cas* gene, then the CRISPR system becomes inert, meaning there will be no additional effect from subsequent insertions into the arrays. CRISPR arrays may also be more recombinogenic due to the repeat sequences, which might influence IS stability. Why not compare overall IS rate across the whole genome, or per kb?

With that said, this does not diminish the functional implications of insertions into *cas* genes. I agree with the overall conclusions that such ISs can balance the costs and benefits of MGEs. Adding this distinction to the discussion would probably be sufficient.

Response: We sincerely appreciate the reviewer's insightful comment regarding our choice to analyze ISs transpositions into *cas* genes alongside CRISPR arrays. We selected CRISPR arrays as a suitable comparative target, primarily because of their relatively similar occurrence frequencies to *cas* genes⁴. Our initial analysis uncovered a distinct predilection for IS transpositions, exclusively occurring within *cas* genes, while no IS insertions were detected within the CRISPR arrays. However, as mentioned by the reviewer, this analytical method may not be sufficiently thorough. Nevertheless, the exclusive preferential transpositions of ISs into *cas* genes and the absence of IS insertions within the CRISPR arrays indicate a significant disparity in statistical probability. This distinction, to a certain extent, implies that ISs transpositions may not be completely arbitrary.

To comprehensively evaluate the observed discrepancy, we endeavored to conduct a theoretical analysis, as recommended by the reviewer, comparing the IS transposition rate across the entire genome and normalizing it on a length scale. We therefore formulated a series of mathematical equations as follows, to quantify the discrepancy in IS transposition rates between the *cas* and non-*cas* genes on a length scale.

$$P_{cas} = nIS_{cas} \frac{\sum_{i=1}^n Len_{cas_i}}{Len_{genome}} \quad (1)$$

$$P_{other} = nIS_{other} \frac{Len_{genome} - \sum_{i=1}^n Len_{cas_i}}{Len_{genome}} \quad (2)$$

$$FC_{Len} = \frac{Len_{genome} - \sum_{i=1}^n Len_{cas_i}}{\sum_{i=1}^n Len_{cas_i}} \quad (3)$$

$$FC_P = \frac{P_{other}}{P_{cas}} \quad (4)$$

Equation (1) calculates the IS transposition rate in all *cas* genes, denoted as P_{cas} . It is determined by multiplying the number of IS insertions into *cas* genes (nIS_{cas}) by the sum of the lengths of all *cas* genes ($\sum_{i=1}^n Len_{cas_i}$) and dividing it by the length of the genome (Len_{genome}).

Equation (2) determines the IS transposition rate in non-*cas* genes, denoted as P_{other} . It is obtained by multiplying the number of IS insertions into non-*cas* genes (nIS_{other}) by the difference between the genome length and the sum of the lengths of all *cas* genes, divided by the length of the genome.

Equation (3) calculates the fold change in gene length, denoted as FC_{Len} . It is obtained by dividing the difference between the length of the genome and the sum of the lengths of all *cas* genes by the sum of the lengths of all *cas* genes.

Equation (4) determines the fold change in transposition rate, denoted as FC_P . It is obtained by dividing the IS transposition rate in non-*cas* genes by the IS transposition rate in all *cas* genes.

Based on the established formulas, we can gain insights into the relationship between sequence length, transposition rates and the presence of preferences or biases (how intentional or specific these insertions are) in IS transposition across the genome. Specifically, when there is a substantial disparity between the values of FC_{Len} and FC_P , it suggests that sequence length does not primarily influence the rate of IS transposition across the entire genome, potentially implying that IS transposition may not be random but rather exhibit a certain preference (put simply, if IS transposition is not biased, the longer potential target sequences ought to possess a higher likelihood of IS transposition occurrence). Conversely, if there is minimal difference between FC_{Len}

and FC_P values, it indicates that sequence length indeed predominantly impacts the over rate of IS transposition, potentially implying that IS transposition may lack a specific bias and instead exhibit a more random pattern.

However, we encountered certain constraints during the practical analysis when evaluating the transposition rates of ISs across the entire genome. Firstly, considering the abundance of necessary genetic elements within bacterial chromosomes, any detrimental transposition of ISs into essential genes would lead to microbial lethality⁵, indicating that ISs are not expected to transpose and disrupt essential genes. In light of this, the inadvertent influence of the sum of lengths of numerous essential genes on the actual FC_{Len} and FC_P values should be duly acknowledged. Undoubtedly, a straightforward and effective approach would entail extracting the entirety of essential genes from each genome, focusing solely on the analysis of non-essential genes. Currently, the thorough characterization of essential genes still remains challenging⁶, thus impeding the accurate and efficient analysis of non-essential genes within each genome. Secondly, the existing ISs detection software is limited to determining whether ISs are present or not, falling short in accurately differentiating between ISs that result from transposition events and those that are inherently present. This limitation will also hamper the precise evaluation of the nIS_{cas} and nIS_{other} values in the equations (1) and (2), respectively, thereby impacting the overall analytical outcomes. In summary, our theoretical analysis reveals that the utilization of this approach, which entails the analysis of the transposition rates of ISs at the genomic level in order to infer their randomness or preference, may also exhibit inherent limitations. Nonetheless, we believe that with ongoing efforts to overcome these limitations in future research, this analytical approach holds great promise.

References

1. Mahillon, J. & Chandler, M. Insertion sequences. *Microbiol Mol Biol Rev.* **62**, 725-774 (1998).
2. Shee, C., Gibson, J.L., Darrow, M.C., Gonzalez, C. & Rosenberg, S.M. Impact

- of a stress-inducible switch to mutagenic repair of DNA breaks on mutation in *Escherichia coli*. *Proc Natl Acad Sci U S A*. **108**, 13659-13664 (2011).
3. Henrikus, S.S. *et al.* Single-molecule live-cell imaging reveals RecB-dependent function of DNA polymerase IV in double strand break repair. *Nucleic Acids Res.* **48**, 8490-8508 (2020).
 4. Charpentier, E., Richter, H., van der Oost, J. & White, M.F. Biogenesis pathways of RNA guides in archaeal and bacterial CRISPR-Cas adaptive immunity. *FEMS Microbiol Rev.* **39**, 428-441 (2015).
 5. Charbonneau, A.R.L. *et al.* Defining the ABC of gene essentiality in *streptococci*. *BMC Genomics.* **18**, 426 (2017).
 6. Chen, A. & Seifert, H.S. Saturating mutagenesis of an essential gene: a majority of the *Neisseria gonorrhoeae* major outer membrane porin (PorB) is mutable. *J Bacteriol.* **196**, 540-547 (2014).